

# Measuring acetylene with a cavity ring-down spectroscopy gas analyser and its use as a tracer to quantify methane emissions

Adil Shah[1], Olivier Laurent[1], Pramod Kumar[1], Grégoire Broquet[1], Loïc Loigerot[1], Timothé Depelchin[2], Mathis Lozano[1], Camille Yver Kwok[1], Carole Philippon[1], Clément Romand[2], Elisa Allegrini[2], Matthieu Trombetti[2] and Philippe Ciais[1]

[1]Laboratoire des Sciences du Climat et de l'Environnement (CEA-CNRS-UVSQ), Institut Pierre-Simon Laplace, Université Paris-Saclay, Site de l'Orme des Merisiers, 91191 Gif-sur-Yvette, France
[2]SUEZ Air and Climate, 15-27 Rue de Port, 92000 Nanterre, France

*Correspondence to*: Adil Shah (adil.shah@lsce.ipsl.fr)

**Abstract.** Facility-scale methane emission fluxes can be derived by comparing tracer and methane mole fraction measurements downwind of a methane emission source where a co-located tracer gas is released at a known flux rate. Acetylene is a commonly used tracer for methane due to its availability, low cost and low atmospheric background. Acetylene mole fraction can be measured using infrared gas analysers such as the Picarro G2203, using cavity ring-down spectroscopy. However, failure to calibrate tracer gas analysers may influence methane flux estimation, if raw mole fraction measurements diverge from their true levels. We conducted extensive Picarro G2203 laboratory characterisation testing. Picarro G2203 acetylene measurements were calibrated by diluting a high concentration of acetylene with ambient air. In order to determine the precise level of acetylene in each calibration gas mixture, a high concentration methane source was diluted in an identical way, with reliable methane mole fraction measurements used to quantify the true level of dilution. It was found that raw Picarro G2203 acetylene mole fraction measurements could be corrected through direct multiplication with a calibration gain factor of 0.94, derived by applying a linear fit between raw measured and reference acetylene mole fraction. However, this calibration is only valid from an acetylene mole fraction of 1.16 ppb, below which unstable measurements were observed by the Picarro G2203 tested in this study. A field study was then conducted by performing fourteen successful transects downwind of an active landfill site, where acetylene was released from a single point location at a fixed flow rate. Methane fluxes were derived by integrating the methane and acetylene mole fraction plumes, as a function of distance along the sampling road. This resulted in a flux variability of 56% between methane flux estimates from different transects which was principally due to flux errors associated with the tracer release location and downwind sampling positioning. Methane fluxes were also derived using raw uncalibrated Picarro G2203 acetylene mole fraction measurements instead of calibrated measurements, which resulted an average methane emission flux underestimation of 7.6%, compared to fluxes derived using calibrated measurements. Unlike a random uncertainty, this 7.6% bias represents a consistent flux underestimation that cannot be reduced with improvements to the field sampling methodology. This study therefore emphasises the equal importance of calibrating both target as well as tracer gas measurements, regardless of the instrument being used to obtain these measurements. Otherwise, biases can be





induced within target gas flux estimates. For the example of methane, this can influence our understanding of the role of certain facility scale emissions within the global methane budget.

## 1. Introduction

The greenhouse effect, which was first proposed over a century ago, is responsible for elevating Earth's near-surface temperature (Arrhenius, 1896). It is caused by various atmospheric species of which methane is the third most important (Mitchell, 1989). Methane had an effective radiative forcing of $0.54 \mathrm{~W~m}^{-2}$ in 2019, which was one quarter that of carbon dioxide (Dentener *et al.*, 2021). Methane has an annualised average background mole fraction of greater than 1.9 ppm (Dlugokencky *et al.*, 1994, Lan *et al.*, 2024), which is over twice as high as it has ever been up to 800 000 years prior to the

onset of the industrial era, defined here as the year 1850 (Chappellaz *et al.*, 1990, Etheridge *et al.*, 1998, Loulergue *et al.*, 2008). Recent estimates suggest that anthropogenic emissions may be responsible for over 50% of total methane emissions (Saunois *et al.*, 2020). Yet there remain large uncertainties in the global methane budget (Dlugokencky *et al.*, 2011, Kirschke *et al.*, 2013, Lan *et al.*, 2021), which is in large part due to uncertainties in emissions from anthropogenic facility-scale sources (Jackson *et al.*, 2020, Bastviken *et al.*, 2022) such as oil and gas extraction infrastructure (Foulds *et al.*, 2022, Wang *et al.*,

2022), agricultural facilitates (Shah *et al.*, 2020, Hayek and Miller, 2021, Marklein *et al.*, 2021), wastewater treatment facilities (Moore *et al.*, 2023, Song *et al.*, 2023) and landfill sites (Maasakkers *et al.*, 2022, Kumar *et al.*, 2024, Wang *et al.*, 2024). Saunois *et al.* (2020) estimated that landfills and waste collectively contributed towards approximately 9% of total methane emissions in the year 2017.

Emissions from individual facility-scale methane sources can be quantified either using inventory-based bottom-up methods

or atmospheric measurement-based top-down methods (Chen and Prinn, 2006, Nisbet and Weiss, 2010, Alvarez *et al.*, 2018, Desjardins *et al.*, 2018, Vaughn *et al.*, 2018). Bottom-up fluxes typically multiply a quantitative activity factor (representative of the amount of an emitting activity taking place) by a qualitative emission factor (Saunois *et al.*, 2016, Allen *et al.*, 2022). These bottom-up emission factors may be more general values assigned for a geographical region (Scarpelli *et al.*, 2020, Bai *et al.*, 2023), developed from knowledge of source emissions (Wolf *et al.*, 2017, Lin *et al.*, 2021) or informed by process

models (Scheutz *et al.*, 2009, Stavert *et al.*, 2021). Meanwhile, most top-down emission estimates rely on atmospheric methane measurements combined with wind data to infer emissions within an inversion model (Denmead *et al.*, 2000, Ars *et al.*, 2017, Cusworth *et al.*, 2021). Top-down facility scale methane emission flux ($Q_{\mathrm{methane}}$) estimates are essential to complement, improve and verify corresponding bottom-up estimates (Guha *et al.*, 2020, Delre *et al.*, 2017, Hayek and Miller, 2021, Marklein *et al.*, 2021, Johnson *et al.*, 2023).

Various approaches can be employed to derive top-down $Q_{\mathrm{methane}}$ (Johnson and Johnson, 1995, Bastviken *et al.*, 2022). Some methods use remote sensing sampling, where mole fraction measurements are integrated over a certain distance (Cusworth *et al.*, 2020, Hrad *et al.*, 2021, Maasakkers *et al.*, 2022, Cossel *et al.*, 2023). However, here we focus on methods using *in situ*



sampling, which provides methane mole fraction ($[CH_4]$) measurements at the sampling point (Feitz *et al.*, 2018). There are many ways to derive top-down $Q_\text{methane}$ by applying an extensive variety of inversion dispersion methods to downwind *in situ* sampling (Flesch *et al.*, 1995, Sonderfeld *et al.*, 2017, Hrad *et al.*, 2021, Shaw *et al.*, 2021, Liu *et al.*, 2024). Rather than using measurements from a single location, downwind sampling transects can be used in more accurate flux methods. For example, both one-dimensional (Foster-Wittig *et al.*, 2015, Yacovitch *et al.*, 2015, Albertson *et al.*, 2016, Riddick *et al.*, 2020, Kumar *et al.*, 2021, Kumar *et al.*, 2022) and two-dimensional (Lee *et al.*, 2018, Shah *et al.*, 2019) downwind transects can be used within Gaussian plume modelling, while mass-balance box modelling can be applied to two-dimensional downwind sampling (Denmead *et al.*, 1998, Foulds *et al.*, 2022, Pühl *et al.*, 2024).

As well as simply using downwind $[CH_4]$ *in situ* sampling, top-down $Q_\text{methane}$ can be derived using a tracer gas release, where the release of a carefully controlled quantity of a tracer gas is co-located with the methane emission source (Lamb *et al.*, 1995, Czepiel *et al.*, 1996). Tracer-based $Q_\text{methane}$ estimation relies on simultaneous *in situ* downwind measurements of both $[CH_4]$ and the tracer gas mole fraction, with no wind measurements required (Johnson and Johnson, 1995, Mønster *et al.*, 2014, Ars *et al.*, 2017). The ratio between the enhancement (above background levels) of the mole fractions of the two gases can be used to yield $Q_\text{methane}$ by direct multiplication with the known tracer release rate (Yacovitch *et al.*, 2017, Feitz *et al.*, 2018, Mønster *et al.*, 2019). A more accurate $Q_\text{methane}$ estimate can be derived by taking the ratio between integrated downwind methane and tracer gas plumes (Scheutz *et al.*, 2011, Mønster *et al.*, 2015, Yver Kwok *et al.*, 2015), which is especially useful to minimise errors due to suboptimal acetylene release co-location with the methane emission source (Ars *et al.*, 2017). Although this manuscript focuses on methane as the target gas of interest, the same principles and outcomes apply to any other target gas whose emission flux is derived using a tracer gas release.

The well-established tracer-based flux method has widely been considered to be a more accurate top-down flux quantification approach for localised facility scale sources, against which to test other methods, although good execution of the method (*i.e.* suitable tracer release location and downwind sampling location) is essential for optimal $Q_\text{methane}$ estimation (Yver Kwok *et al.*, 2015, Bell *et al.*, 2017, Mønster *et al.*, 2019, Song *et al.*, 2023). The accuracy of tracer-based fluxes has been confirmed during various controlled release experiments using co-located methane and tracer point source releases (Mønster *et al.*, 2014, Feitz *et al.*, 2018, Liu *et al.*, 2024), but also with an offset tracer and methane source, which can result in greater flux uncertainty (Ars *et al.*, 2017, Fredenslund *et al.*, 2019). Raw tracer-based $Q_\text{methane}$ has been derived in countless previous studies from a multitude of facility-scale methane sources, for example from cattle (Johnson *et al.*, 1994, Daube *et al.*, 2019, Vechi *et al.*, 2022), oil and gas extraction infrastructure (Lamb *et al.*, 1995, Omara *et al.*, 2016, Yacovitch *et al.*, 2017), anaerobic digesters (Scheutz and Fredenslund, 2019), wastewater treatment facilities (Yver Kwok *et al.*, 2015, Delre *et al.*, 2017, Delre *et al.*, 2018) and landfill sites (Czepiel *et al.*, 1996, Galle *et al.*, 2001, Mønster *et al.*, 2015, Matacchiera *et al.*, 2019, Mønster *et al.*, 2019).



The choice of tracer gas for methane has also evolved (Scheutz *et al.*, 2009, Delre *et al.*, 2018, Mønster *et al.*, 2019). Originally, sulphur hexafluoride was favoured (Johnson and Johnson, 1995, Czepiel *et al.*, 1996) due to its inert properties and almost absent atmospheric background. However, sulphur hexafluoride is an incredibly potent greenhouse gas with an atmospheric lifetime of roughly 1 000 years (Kovács *et al.*, 2017). Carbon dioxide has also been used as a tracer gas (Lamb *et al.*, 1995), but it has extensive background variability due to a multitude of localised sources and sinks (Grimmond *et al.*, 2002, Schwandner *et al.*, 2017). Nitrous oxide is a more recent alternative tracer gas option to quantify methane emissions (Galle *et al.*, 2001, Mønster *et al.*, 2015, Omara *et al.*, 2016). Though this is a potent greenhouse gas, it has a finite atmospheric lifetime due to its reaction with atmospheric oxygen radicals and due to soil consumption (Cicerone, 1989, Kroeze, 1994, Tian *et al.*, 2020). Finally, acetylene has more recently been used as a tracer for methane (Yver Kwok *et al.*, 2015, Fredenslund *et al.*, 2019, Scheutz and Fredenslund, 2019, Vechi *et al.*, 2022). It readily reacts with the hydroxyl radical, resulting in a relatively short lifetime of up to a month (Kanakidou *et al.*, 1988, Gupta *et al.*, 1998, Hopkins *et al.*, 2002, Crounse *et al.*, 2009). Acetylene is also cheap to produce. However, it has a flammable atmospheric range of between 2.5% and approximately 80% (Williams and Smith, 1969), which is the largest range of any readily available gas. Nevertheless, background levels of no greater than 1 ppb (Kanakidou *et al.*, 1988, Gupta *et al.*, 1998, Hopkins *et al.*, 2002, Xiao *et al.*, 2007), makes it a preferred option compared to nitrous oxide, which can otherwise be emitted from many sources including agricultural activities, resulting in a variable atmospheric nitrous oxide background (Tian *et al.*, 2020).

The ability to obtain *in situ* measurements of both methane and the chosen tracer gas underpins the accuracy of any derived tracer-based flux. Acetylene mole fraction ($[C_2H_2]$) has traditionally been measured using flame-ionisation gas chromatography (Kanakidou *et al.*, 1988, Hopkins *et al.*, 2002, Crounse *et al.*, 2009) and Fourier-transform infrared (IR) spectroscopy (Xiao *et al.*, 2007, Feitz *et al.*, 2018), which are both slow *in situ* techniques. The recent use of acetylene as methane tracer has largely been facilitated by the development of fast-response (less than 1-minute sampling frequency) *in situ* measurement techniques. Yacovitch *et al.* (2017) derived tracer-based fluxes with a sensor using direct IR spectroscopy with a tuneable laser, manufactured by Aerodyne Research, Inc. (Billerica, Massachusetts, USA), with a detection limit of 78 ppt and a $[C_2H_2]$ linear calibration uncertainty of 3% (assuming a zero intercept). This calibration was performed by diluting gas from an acetylene cylinder, although the $[C_2H_2]$ testing range is not provided (Yacovitch *et al.*, 2017). The Ultraportable Methane-Acetylene Analyzer (ABB Ltd, Zürich, Switzerland) has also been used to measure $[C_2H_2]$, which uses off-axis integrated cavity output spectroscopy, with a manufacturer-rated precision of less than 1 ppb at 0.2 Hz (Fredenslund *et al.*, 2019). Feitz *et al.* (2018) tested this instrument to verify for linearity using two cylinders containing 4.1 ppb $[C_2H_2]$ and 20.6 ppm $[C_2H_2]$, although without providing correlation results.

The Picarro G2203 (Picarro, Inc., Santa Clara, California, USA) is one of the most widely used acetylene gas analysers, which has been operated in numerous tracer release studies (Mønster *et al.*, 2015, Yver Kwok *et al.*, 2015, Ars *et al.*, 2017, Delre *et al.*, 2017, Delre *et al.*, 2018, Vechi *et al.*, 2022). It uses cavity ring-down spectroscopy (CRDS) to detect small $[C_2H_2]$



enhancements of less than 1 ppb (Mønster *et al.*, 2014). The Picarro G2203 has a manufacturer-rated precision of less than $\pm 0.6$ ppb at 0.5 Hz (Picarro, Inc., 2015). Mønster *et al.* (2014) provided a brief testing overview of the Picarro G1203 (which is spectroscopically similar to the Picarro G2203, but with older electronics) using a testing cylinder containing 103 ppb $[C_2H_2]$ in synthetic air, with a 10% $[C_2H_2]$ uncertainty. However, they provided limited details on their characterisation testing procedure, such as whether the gas was diluted to sample lower $[C_2H_2]$ levels and the number of sampling steps, if any (Mønster *et al.*, 2014). In a tracer release study by Omara *et al.* (2016), regular Picarro G2203 calibrations were conducted using a single 100 ppb $[C_2H_2]$ gas standard to check for drift. They measured a raw acetylene mole fraction ($[C_2H_2]_r$) of $(112 \pm 3.2)$ ppb (Omara *et al.*, 2016); this +12% error emphasises the risk in using raw measurements from tracer gas analysers.

To summarise, there exists a large body of research having used tracer-based methods to estimate $Q_{methane}$, but a lack of emphasis in calibrating the tracer gas measurement in these previous studies. It is vitally important to conduct independent rigorous testing of gas analysers such as the Picarro G2203, across the full $[C_2H_2]$ range expected during field sampling. This is essential due to the reliance of tracer-based $Q_{methane}$ estimates to inform site operators and policy makers. Any disparity in $[C_2H_2]$ measurements may be projected as persistent biases in $Q_{methane}$ estimates, emphasising the key importance of this work.

We provide here the first detailed characterisation, to our knowledge, of the Picarro G2203 gas analyser for measuring $[C_2H_2]$ in **Sect. 2**, including the influence of water. We describe the implementation of the gas analyser to conduct an acetylene release from a landfill site in France in **Sect. 3**. We also present a comprehensive description of the equipment used within our acetylene release method in **Sect. 3**. The purpose of this study is not to evaluate emissions from this specific landfill site in the context of methane emissions compared to other sources, but rather to focus on the tracer-based flux quantification method itself. In this study, the chosen landfill site serves only as a complex heterogeneous test site with which to test our methods and the specificities of this particular site are beyond the scope of this work. Flux results from this study site are presented in **Sect. 4**, where we discuss the variability in $Q_{methane}$ results and disparity between $Q_{methane}$ values derived using raw versus calibrated mole fraction measurements. We summarise the implications on $Q_{methane}$ quantification of using raw mole fraction measurements without applying an acetylene calibration in **Sect. 5**.

## 2 Instrument acetylene response characterisation

### 2.1 Testing equipment

The Picarro G2203 gas analyser uses CRDS to measure $[C_2H_2]_r$, raw water mole fraction ($[H_2O]_r$) and raw methane mole fraction ($[CH_4]_r$) This section is dedicated to characterising Picarro G2203 $[C_2H_2]_r$, measurements, with a Picarro G2203 $[CH_4]_r$ measurement calibration provided in **Sect. S1** in the **Supplement**. A Picarro G2401 (Picarro, Inc.) gas analyser was also used during this Picarro G2203 characterisation work, which also measures $[H_2O]_r$ and $[CH_4]_r$, but not $[C_2H_2]_r$. The CRDS method used by the Picarro G2230 and Picarro G2401 gas analysers derives mole fraction measurements using a spectrum of the





characteristic exponential decay "ring-down" time of IR radiation leaking out of a cavity (Paldus and Kachanov, 2005) held under controlled pressure and temperature (Crosson, 2008). IR radiation from a tuneable distributing feedback laser is injected into the cavity at discrete points across a narrow wavelength range, which is tuned to the absorption peak of interest (Crosson, 2008). IR absorption occurs in the cavity following the Beer-Lambert Law at absorbing wavelengths (Lambert, 1760).

Following laser build-up, the laser is switched off and radiation leaks out of the cavity (Paldus and Kachanov, 2005). The ring-down times of leaking radiation are used to produce an absorbance spectrum as a function of wavelength. The ratio between the maximum absorbance signal and the signal at a baseline wavelength (representative of sampling in an empty cavity) is used to derive gas mole fraction using internal instrumental algorithms.

Throughout each laboratory test conducted during this work, the Picarro G2203 and Picarro G2401 were connected in parallel.

Both the Picarro G2203 and Picarro G2401 record raw mole fraction measurements for each gas individually, each with a unique timestamp. Therefore, all Picarro G2401 measurements were shifted to the Picarro G2203 timestamp, by applying a lag time correction. At the time of testing, Picarro G2203 $[C_2H_2]_r$ measurements had an average observed sampling frequency of $(0.3\pm0.1)$ Hz in dry conditions. The time interval between each measurement for the Picarro G2203 used in this study followed a roughly 40 s periodic cycle every ten measurements of roughly 2 s, 4 s, 2 s, 4 s, 2 s, 4 s, 2 s, 4 s, 2 s and 13 s

between measurements. All $[C_2H_2]_r$ and $[CH_4]_r$ measurements presented in this manuscript are defined as wet measurements, with no internal instrumental water correction applied to this data.

During testing, a specially prepared 20 dm³ acetylene calibration cylinder was used (Air Products N.V., Diegem, Belgium). This was volumetrically filled with an $[C_2H_2]$ of 10 180 ppb in argon with a $\pm3\%$ uncertainty, according to the cylinder provider. This high $[C_2H_2]$ level was chosen to allow for high levels of dilution, to minimise the effect of the argon in this

cylinder on the natural balance of air; changes in air composition can affect spectral fitting by changing the shape of IR absorption peaks (Lim *et al.*, 2007, Rella *et al.*, 2013). A 20 dm³ methane calibration cylinder (Air Products N.V.) was also used during testing. This was gravimetrically filled with a $[CH_4]$ of 995.4 ppm in argon with a $\pm0.5\%$ uncertainty, according to the cylinder provider. Dilution of these two calibration cylinders was performed using gas from three cylinders containing natural ambient compressed outside air, assumed to contain a background acetylene mole fraction ($[C_2H_2]_0$) level of 0 ppb (due

to the absence of nearby acetylene sources).

All tests were conducted using mass-flow controller (MFC) units (EL-FLOW Select, Bronkhorst High-Tech B. V., AK Ruurlo, Netherlands), which were used to generate gas blends and to control gas flow. All laboratory testing was conducted using either stainless-steel (SS) tubing or Synflex 1300 tubing (Eaton Corporation plc, Dublin, Ireland) with an outer diameter (OD) of 0.25 inches, in conjunction with standard SS Swagelok fittings (Swagelok Company, Solon, Ohio, USA) which were used

to connect tubing and various components. The gas stream was filtered using 2 μm particle filters (SS-4FW-2, Swagelok Company) to protect downstream instrumentation. One of either two diaphragm pumps was used during testing to pressurise the gas stream: the N86KN.18 (KNF DAC GmbH, Hamburg, Germany) has fittings compatible with Swagelok fittings whereas





the 1410VD/12VDC (Gardner Denver Thomas GmbH, Fürstenfeldbruck, Germany) has barbed fittings which were connected to short lengths of Tygon S3 E-3603 tubing (Saint-Gobain Performance Plastics, Inc., Solon, Ohio, USA) to which Swagelok

fittings were attached. A needle valve (SS-4MG, Swagelok Company) was used to stabilise and restrict the pressure downstream of the diaphragm pumps. A check valve (SS-4C-1, Swagelok Company) was also used to direct gas flow during testing.

As water vapour is naturally present in air, the $[C_2H_2]_r$ response of the Picarro G2203 was tested under various water mole fraction ($[H_2O]$) levels which could be controlled using three different methods. Water could be added to the gas stream using

a dew-point generator (LI-610, LI-COR, Inc., Lincoln, Nebraska, USA) to saturate passing gas to a fixed dew-point setting. This was incorporated into the gas stream by connecting standard plastic Swagelok fittings to the standard Bev-A-Line IV tubing (Thermoplastic Processes Inc, Georgetown, Delaware, USA) used by the dew-point generator, which has an OD of 0.25 inches and an inner diameter of 0.125 inches. The internal pump of the dew-point generator was by-passed by cutting and adding a standard plastic Swagelok fitting to the Bev-A-Line IV tubing on the instrument labelled as "to condenser". A three-

way ball valve (B-42XS4, Swagelok Company) was placed both upstream and downstream of the dew-point generator; these were used to direct the instrument away from the gas line and towards a direct vent to the atmosphere (*i.e.* with no connection) when adding water to the condenser, to ensure an atmospheric pressure both upstream and downstream of the dew-point generator. Conversely water could be removed from the gas stream using a Nafion-based gas dryer (MD-070-144S-4, Perma Pure LLC, Lakewood, New Jersey, USA), which contains a Nafion membrane (The Chemours Company FC, LLC,

Wilmington, Delaware, USA), which reduced observed Picarro G2203 $[H_2O]_r$ measurements to less than 0.1%. The Nafion-based gas dryer was connected in reflux mode during testing, whereby the gas dryer was placed between the Picarro G2203 and its downstream vacuum pump, to create a vacuum outside the Nafion membrane through which the sample gas passed, as described in detail by Welp *et al.* (2013). Water could be dried further through chemical absorption using magnesium perchlorate grains (ThermoFisher (Kandel) GmbH, Kandel, Germany) in a water scrubber. The effect of any potential artefacts

of the Nafion-based gas dryer, the magnesium perchlorate scrubber and the dew-point generator was tested in **Sect. S2** in the **Supplement**, which showed no significant effect on $[C_2H_2]_r$ response. It was especially important to verify this for the dew-point generator which bubbles gas through a water reservoir, as acetylene has a solubility in water of 1.1 g dm$^{-3}$ at 20° C (Priestley and Schwarz, 1940).

## 2.2. Water tests

Water can potentially affect IR mole fraction measurements of any gas due to three key reasons, as described by Rella *et al.* (2013). Two of these reasons are specifically due to IR spectroscopy, although the magnitude and importance of each of these effects depends on the specific measured gas in question. Firstly, spectral interference can occur where a water IR absorption line overlaps with a target gas absorption line. Secondly, an independent peak broadening effect occurs whereby the shape of the absorption peak for the target gas of interest can change due to interactions with water in the gas mixture, which effect the



dipole of the target gas in question, therefore causing a change in the peak shape. Although any gas can affect the peak shape and cause spectral overlap, the natural balance of air is usually a constant blend of nitrogen, oxygen and argon resulting in a constant effect on the methane spectrum, with water the main variable in ambient air. It is therefore conventional to characterise IR peak shape in the absence of water. Finally, a natural dilution effect occurs (which is not exclusive to IR spectroscopy) where the fraction of target gas molecules that would otherwise be present in dry gas is reduced due to the additional presence of water in the overall gas mixture. It is therefore a standard procedure to convert all gas mole fraction measurements into dry mole fractions as a first step, to which water can then be subsequently reintroduced at a later stage, if required in flux analysis.

An evaluation of the influence of specific spectral effects of water on $[C_2H_2]_r$ measurements is beyond the scope of this study. In this work, the net influence of $[H_2O]$ on $[C_2H_2]_r$ measurements is instead characterised empirically. Preliminary testing was conducted by sampling five different targeted acetylene mole fraction ($[C_2H_2]_t$) levels (6 ppb, 12 ppb, 20 ppb, 30 ppb and 40 ppb) by directly blending gas from the acetylene calibration cylinder with gas from a natural ambient compressed air cylinder. This is illustrated schematically in **Fig. 1** for this test, where the check valve was used to avoid back-flow into the dew-point generator when sampling pure dry gas, therefore minimising the mixing of residual wet air with the dry gas stream. Nine different $[H_2O]$ levels were sampled at each $[C_2H_2]_t$ setting. This was achieved by humidifying a portion of gas using the dew-point generator with a 20° C setting and then blending this with dry air from the same original gas stream, passing though the magnesium perchlorate scrubber, to ensure dryness. First, dry air was sampled for 60 minutes before sampling each wet setting for 15 minutes. The results of this test are presented in **Fig. 2**.

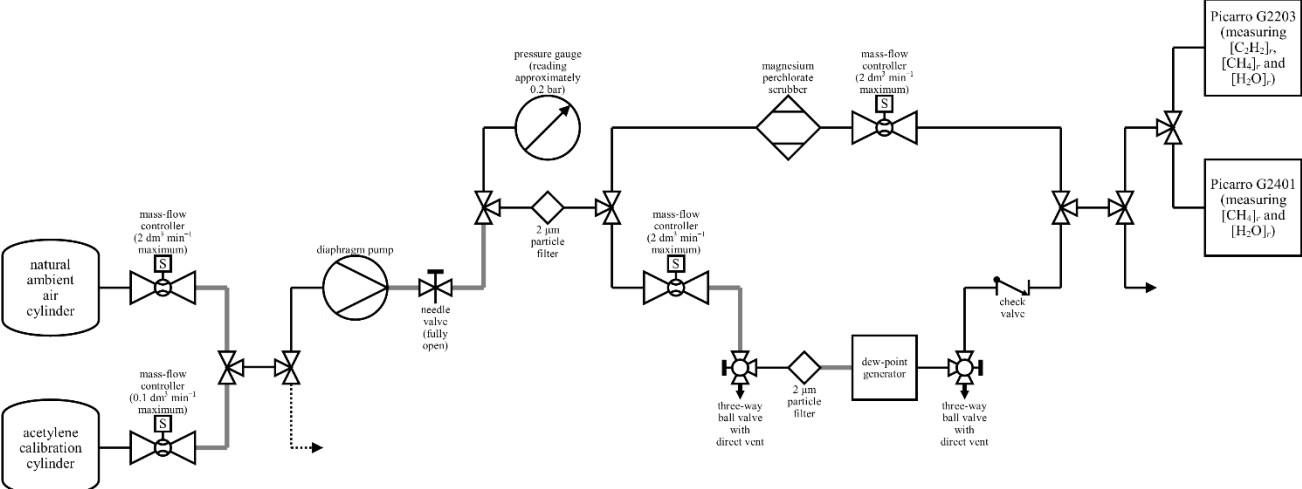

**Figure 1: A schematic of the set-up during water testing. An arrow represents a vent to the atmosphere. Solid black lines represent either SS tubing or Synflex 1300 tubing with an OD of 0.25 inches. Solid grey lines represent SS connections between two components of approximately 0.04 m. The black dashed line represents SS tubing with an OD of 0.125 inches. All connections used standard SS Swagelok fittings. Maximum MFC flow rates are representative of corresponding volumetric flow rates for dry air at 101 325 Pa and 273.15 K. The three-way ball valves were directed towards the gas stream during testing and away from the direct vent to atmosphere.**





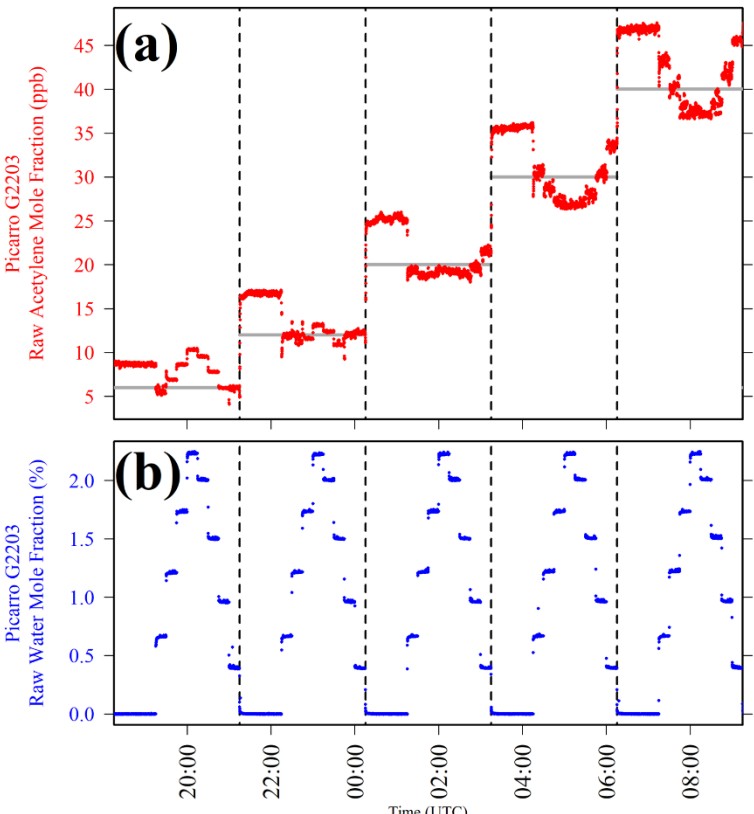

**Figure 2: (a) Picarro G2203 [C₂H₂]ᵣ plotted as red dots and (b) Picarro G2203 [H₂O]ᵣ plotted as blue dots, when sampling nine different [H₂O] levels at five different [C₂H₂]ₜ settings, with the change between each different [C₂H₂]ₜ setting indicated by dashed vertical lines. [C₂H₂]ₜ calculated from MFC settings is plotted in (a) as light grey lines.**

**Figure 2** shows that at each fixed $[C_2H_2]_t$ level (the periods between vertical dashed lines), $[C_2H_2]_r$ as measured by the Picarro G2203 (red dots in **Fig. 2** (a)) changed in response to $[H_2O]$. However, the nature of the $[C_2H_2]_r$ response as a function of increasing $[H_2O]_r$ (blue dots in **Fig. 2** (b)) was not consistent at the different tested $[C_2H_2]_t$ levels (light grey lines in **Fig. 2** (a)) At 6 ppb $[C_2H_2]_t$, $[C_2H_2]_r$ appeared to increase with increasing $[H_2O]_r$. Yet at 12 ppb $[C_2H_2]_t$ and 20 ppb $[C_2H_2]_t$ there was no clear $[H_2O]_r$ relationship. At 30 ppb and 40 ppb $[C_2H_2]_t$, $[C_2H_2]_r$ appeared to decrease with $[H_2O]_r$. While use of a dew-point generator may explain this behaviour (due to solubility of acetylene in the water reservoir (Priestley and Schwarz, 1940)), testing presented in **Sect. S2** in the **Supplement** shows no such effect, making it clear that these effects are due to the gas analyser itself. In addition, $[C_2H_2]_r$ measurements were excessively noisy in the presence of water compared to dry sampling conditions, particularly at higher $[C_2H_2]_t$ levels. The specific cause of these water effects on $[C_2H_2]_r$ (in the context of spectral effects) is beyond the scope of this empirical study. Nevertheless, this result allows us to conclude that noisy $[C_2H_2]_r$ measurements at high $[H_2O]_r$, combined with the inconsistent directions of $[C_2H_2]_r$ changes in response to $[H_2O]_r$ changes suggests that it is not straightforward to derive a reliable simple empirical water correction model across a $[H_2O]$ range typically observed in ambient atmospheric conditions.



Figure 2 shows that $[C_2H_2]_r$ response was most stable (least noisy) at lower $[H_2O]$ levels. Therefore $[C_2H_2]_r$ response to $[H_2O]_r$ was instead tested in dryer conditions. This simulates a $[H_2O]$ range experienced with sole use of the Nafion-based gas dryer, which reduces $[H_2O]_r$ measurements to less than 0.1%. Such a correction could be useful if using the Nafion-based drier to obtain semi-dry gas sampling during eventual field deployment. In this test, the dew-point generator was fixed to a 0° C setting to enable a lower range of $[H_2O]$ levels to be sampled. The same procedure as for the previous test was carried out, but in this test, the entire procedure was performed twice to test for repeatability. The results of this test are presented in **Fig. 3**.

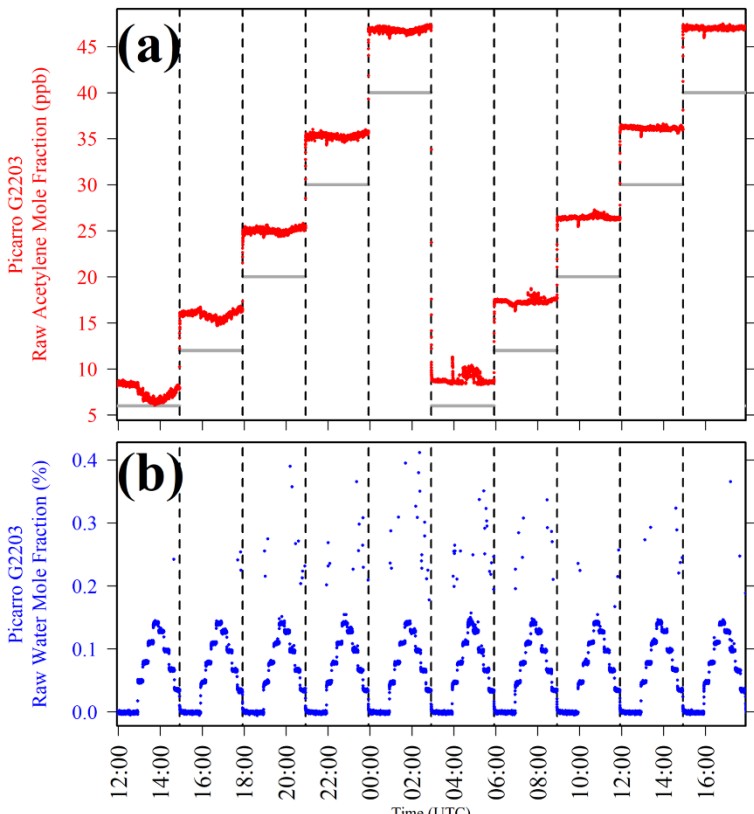

**Figure 3: (a) Picarro G2203 $[C_2H_2]_r$ plotted as red dots and (b) Picarro G2203 $[H_2O]_r$ plotted as blue dots, when sampling nine different low $[H_2O]$ levels at five different $[C_2H_2]_t$ settings over two testing cycles, with the change between each different $[C_2H_2]_t$ setting indicated by dashed vertical lines. $[C_2H_2]_t$ calculated from MFC settings is plotted in (a) as light grey lines.**

**Figure 3** shows that despite limiting $[H_2O]_r$ measurements to mostly below 0.2% (blue dots in **Fig. 3** (b)), $[C_2H_2]_r$ remained unpredictable at each fixed $[C_2H_2]_t$ setting, (although $[C_2H_2]_r$ appeared relatively stable at the highest $[C_2H_2]_t$ levels in this water test). It is particularly concerning that $[C_2H_2]_r$ first decreased with increasing $[H_2O]_r$ at 6% $[C_2H_2]_t$ while the opposite behaviour was observed during the second period at the same $[C_2H_2]_t$ level. At the other $[C_2H_2]_t$ settings, the relationship between $[C_2H_2]_r$ and $[H_2O]_r$ was less obvious. Nevertheless, it is clear that there is some impact on $[C_2H_2]_r$ due to the presence of water. This test also shows that $[H_2O]_r$ occasionally spiked when there was no corresponding $[H_2O]$ spike, which was confirmed by Picarro G2401 $[H_2O]_r$ measurements (see **Sect. S3** in the **Supplement**). This means that in reality, these $[H_2O]_r$





outliers were an artefact of the instrumental Picarro G2203 response. This is probably due to issues in spectral water fitting at low (but non-zero) $[H_2O]_r$ levels. It can therefore be concluded that a reliable and repeatable $[H_2O]_r$ correction is difficult to apply to $[C_2H_2]_r$ as the relationship is too unpredictable, even if limiting $[H_2O]_r$ to below 0.1% with a Nafion-based gas dryer during field deployment. Therefore, optimum Picarro G2203 field sampling requires fully dry conditions. This avoids the complications associated with having to devise a reliable $[H_2O]_r$ correction and also, with having to identify and remove spurious $[H_2O]_r$ spikes. It follows that $[C_2H_2]_r$ response should be calibrated in fully dry conditions.

## 2.3 Calibration gas blending characterisation

In order to calibrate dry Picarro G2203 $[C_2H_2]_r$ measurement response, precise reference $[C_2H_2]$ testing gas mixtures are required. As we had no access to acetylene gas standards with which to calibrate $[C_2H_2]_r$ response, gas from the acetylene calibration cylinder could instead be carefully diluted using precise MFC blends. However, preliminary testing revealed MFC flow rates to be unreliable and offset from their predicted settings, which may be due to MFC contamination (due to particles, debris or oils, for example, getting trapped inside the instrument) or general ageing over time. This was especially concerning at low flow rate settings, compared to the maximum range of each MFC.

To characterise any disparity between the actual $[C_2H_2]$ level in the gas mixture and $[C_2H_2]_t$, an empirical mass-flow controller correction factor ($C_{MFC}$) was derived, with each $C_{MFC}$ value corresponding to a specific set of MFC settings. This factor can be directly applied to the enhancement in $[C_2H_2]_t$ above the $[C_2H_2]_0$ level in the dilution gas where

$$[C_2H_2] = \left( C_{MFC} \cdot \left( [C_2H_2]_t - [C_2H_2]_0 \right) \right) + [C_2H_2]_0, \tag{1}$$

to yield reference $[C_2H_2]$ levels. $C_{MFC}$ values can be derived as a function of each set of MFC settings by comparing targeted and measured levels of a different proxy gas blended with a dilution gas using the same MFC settings. Methane was used as a proxy gas for this purpose following

$$C_{MFC} = \frac{[CH_4] - [CH_4]_0}{[CH_4]_t - [CH_4]_0}, \tag{2}$$

which uses the enhancement in targeted methane mole fraction ($[CH_4]_t$) above the background methane mole fraction ($[CH_4]_0$) level in the dilution gas. Accurate $[CH_4]$ measurements could be obtained here by calibrating Picarro G2401 $[CH_4]_r$ measurements using six certified gas standards traceable to the World Meteorological Organisation (WMO) greenhouse gas scale for methane (WMO X2004A) of between 1.6 ppm and 3.3 ppm $[CH_4]$. This yielded a gain factor of 1.0073 and an offset of $-0.002647$ ppm with a calibration root-mean squared error (RMSE) of $\pm 0.000079$ ppm, for the Picarro G2401 used in this work. This method makes the effect of any specific MFC errors in $[C_2H_2]_t$ and $[CH_4]_t$ estimation redundant, as they cancel out when correcting $[C_2H_2]_t$ using Eq. (1) in conjunction with Eq. (2).



To derive $C_{MFC}$, gas from the methane calibration cylinder was blended with gas from a natural ambient compressed air cylinder, with a $[CH_4]_0$ of 2.057 ppm. Twenty different $[CH_4]_t$ levels were targeted between $[CH_4]_0$ and 11.82 ppm. First, $[CH_4]_0$ (*i.e.* pure natural ambient compressed air) was sampled for 60 minutes before sampling each other $[CH_4]_t$ setting for 15 minutes. This cycle was repeated three times before finally sampling $[CH_4]_0$ for 60 minutes. As the methane calibration

cylinder has a high (995.4 ppm) $[CH_4]$ content, the total quantity of gas from this cylinder reaching the gas analysers was less than 1% (*i.e.* representing a small argon enhancement and thus, causing minimal influence on spectral shape). Low $[CH_4]_t$ levels were obtained through a system of double dilution, as illustrated schematically in **Fig. 4**. First, gas from the methane calibration cylinder was diluted with natural ambient compressed air. This was then subsampled and blended with compressed air a second time.

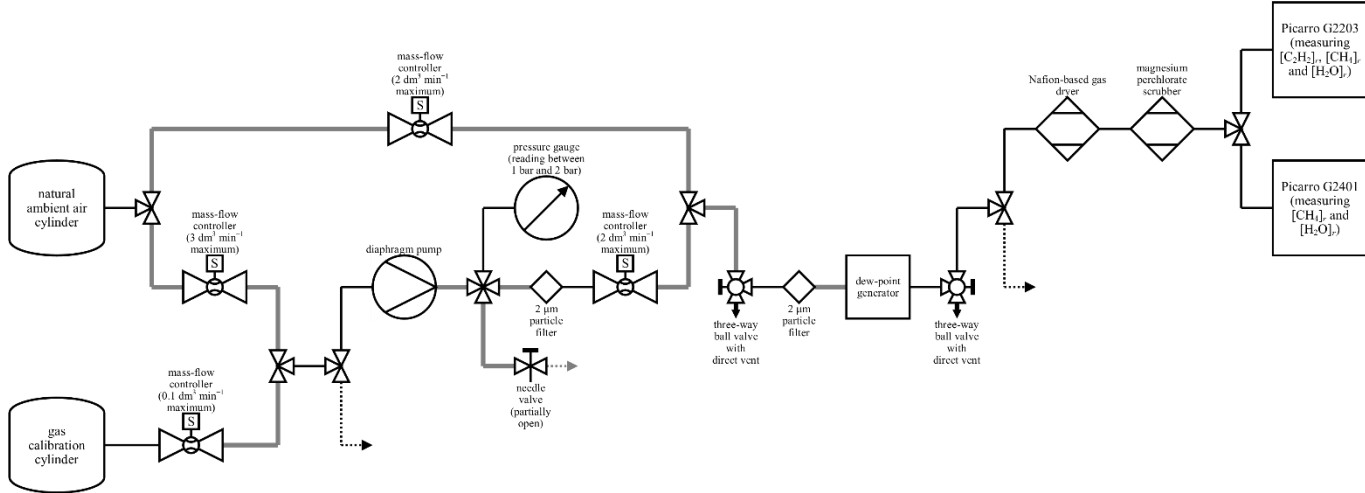

**Figure 4: A schematic of the set-up during acetylene calibration and MFC blending characterisation. An arrow represents a vent to the atmosphere. Solid black lines represent either SS tubing or Synflex 1300 tubing with an OD of 0.25 inches. Solid grey lines represent SS connections between two components of approximately 0.04 m. The black dashed line represents SS tubing with an outer diameter of 0.125 inches. The grey dashed line represents SS tubing with an OD of 0.0625 inches. All connections used standard**

**SS Swagelok fittings. Maximum MFC flow rates are representative of corresponding volumetric flow rates for dry air at 101 325 Pa and 273.15 K. The gas calibration cylinder was the methane calibration cylinder during blending characterisation and the acetylene calibration cylinder during acetylene calibration. The three-way ball valves were directed towards the gas stream during testing and away from the direct vent to atmosphere.**

**Figure 5** shows results for the methane MFC blending characterisation test for the Picarro G2401, where all $[CH_4]_r$

measurements have been converted into $[CH_4]$, using the WMO standard calibration coefficients given above. To derive $C_{MFC}$ values from this data, a 5-minute average Picarro G2401 $[CH_4]$ value was taken from towards the end of each 15-minute sampling step (except when sampling $[CH_4]_0$). This averaging period was used to enable the Picarro G2401 to stabilise and to flush all gas tubing. As the cycle was repeated thrice, each $[CH_4]_t$ level has three corresponding 15-minute $[CH_4]$ averages, of which the average was used within Eq. **(2)** to derive $C_{MFC}$ values, which are plotted in **Fig. 5** (c) as a function of corresponding

$[C_2H_2]_t$ values derived using the same MFC settings (see next subsection for details). The standard deviation of each average (*i.e.* the standard deviation between each of three 5-minute $[CH_4]$ averages at each $[C_2H_2]_t$ level greater than $[CH_4]_0$) was on



average $(\pm 0.002 \pm 0.001)$ ppm. This small variability demonstrates the reliability in MFCs to consistently provide the same gas blends on multiple occasions over time, with a relatively large gap of 5.75 hours between each of the three sampling cycles. In summary **Fig. 5** (c) $C_{\text{MFC}}$ values show that the influence of a $[C_2H_2]$ enhancement above $[C_2H_2]_0$ can be over 200% larger than a corresponding $[C_2H_2]_t$ enhancement (above $[C_2H_2]_0$), emphasising the importance of this MFC blending characterisation approach, as opposed to assuming $[C_2H_2]$ to equal $[C_2H_2]_t$ in the subsequent acetylene calibration analysis.





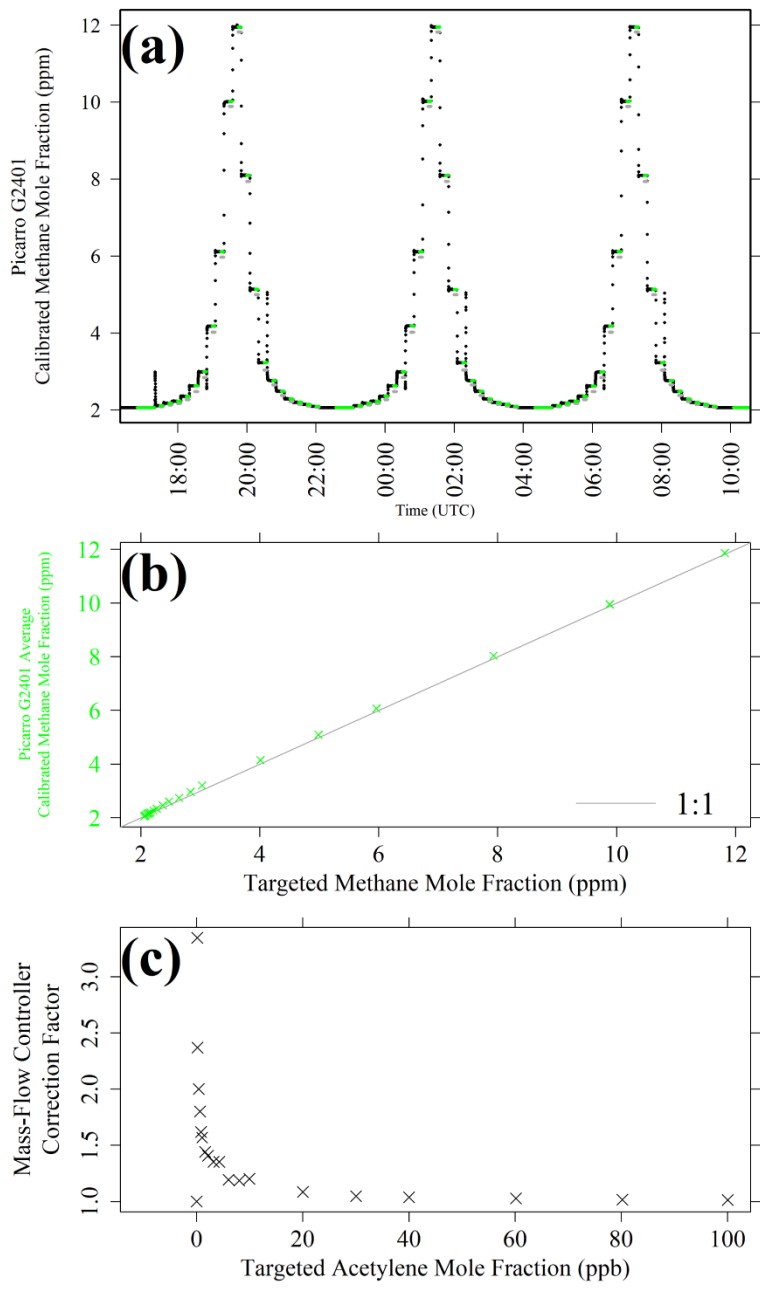

**Figure 5: (a) Picarro G2401 [CH4] plotted as black dots, (b) corresponding [CH4] 5-minute averages plotted as green crosses against [CH4]t calculated from MFC settings and (c) CMFC as a function of corresponding [C2H2]t levels, derived from three testing cycles by blending gas from the methane calibration cylinder with natural ambient compressed air. Periods used to derive averages are highlighted as green dots and corresponding [C$_2$H$_2$]$_t$ levels are shown in the background as light grey dots in (a). An identity line is shown as a solid light grey line in (b).**





## 2.4. Acetylene calibration

Having characterised MFC gas blending capability at specific MFC flow rate settings, $[C_2H_2]_t$ values can be converted into

corresponding $[C_2H_2]$ using Eq. **(1)**. This provides corrected $[C_2H_2]$ gas blend standards with which to calibrate Picarro G2203 $[C_2H_2]_r$ measurements. Gas from the acetylene calibration cylinder was blended with gas from the same natural ambient compressed air cylinder used during MFC blending characterisation, containing 0 ppb $[C_2H_2]_0$. The same process of double dilution was used, as illustrated schematically in **Fig. 4**. Identical flow rate settings to those used during MFC blending characterisation resulted in the sampling of twenty different $[C_2H_2]$ levels between 0 ppb and 101.3 ppb (*i.e.* where each $[C_2H_2]_t$

level is corrected here by its corresponding $C_{MFC}$ value given in **Fig. 5** (c)). First, 0 ppb $[C_2H_2]$ was sampled for 60 minutes before sampling each other $[C_2H_2]$ setting for 15 minutes. Both the MFC blending characterisation test and acetylene calibration test were conducted within a 48-hour window, to minimise drift in MFC performance, and with no MFC power loss. This $[C_2H_2]$ range is deemed to be sufficient to capture most $[C_2H_2]$ measurements typically expected to be measured downwind of a controlled acetylene release, although a larger calibration range may be required if sampling nearer to the

source, where higher $[C_2H_2]$ sampling may be expected.

Ordinarily, gas from compressed cylinders is already dry, so no dying is required. However, during this test (as well as during the blending characterisation test above), all gas passed through the dew-point generator with a 8° C setting, to humidify the gas stream. The gas then passed though the Nafion-based gas dryer to significantly reduce $[H_2O]$ before finally passing through the magnesium perchlorate scrubber, to ensure dryness. This counterintuitive procedure of humidification followed by drying

was used to best replicate sampling in the field when using both the Nafion-based gas dryer as well as the magnesium perchlorate scrubber, to account for potential artefacts on $[C_2H_2]$ at the point of measurement. Although a dew-point generator is not present during field sampling, **Sect. S2** in the **Supplement** shows that this has no noticeable effect on $[C_2H_2]_r$ measurements, alongside the Nafion-based gas dryer and the magnesium perchlorate scrubber. Nevertheless, it was still preferred to carry out this humidification and drying procedure as an added precaution.

Sampling results are presented in **Fig. 6** for the acetylene calibration test for the Picarro G2203. **Figure 6** (b) shows a stable Picarro G2203 $[H_2O]_r$ level throughout testing, as expected. A calibration could be derived from this data by taking a 5-minute average Picarro G2203 $[C_2H_2]_r$ value from towards the end of each 15-minute sampling step. However, for each 60-minute $[C_2H_2]_0$ sampling period, a 30-minute average was used, as $[C_2H_2]_r$ measurements are slightly more noisy at 0 ppb $[C_2H_2]$. For the lowest three non-zero $[C_2H_2]$ settings (0.349 ppb, 0.464 ppb and 0.867 ppb), unstable $[C_2H_2]_r$ measurements were

observed, with $[C_2H_2]_r$ occasionally resolving to the $[C_2H_2]_r$ level observed at 0 ppb $[C_2H_2]$ (see **Sect. S4** in the **Supplement** for an example), despite the fact that the same gas stream was being sampled. This probably corresponds to the Picarro G2203 temporarily losing the acetylene IR absorption peak. Therefore a calibration was derived excluding these data points.



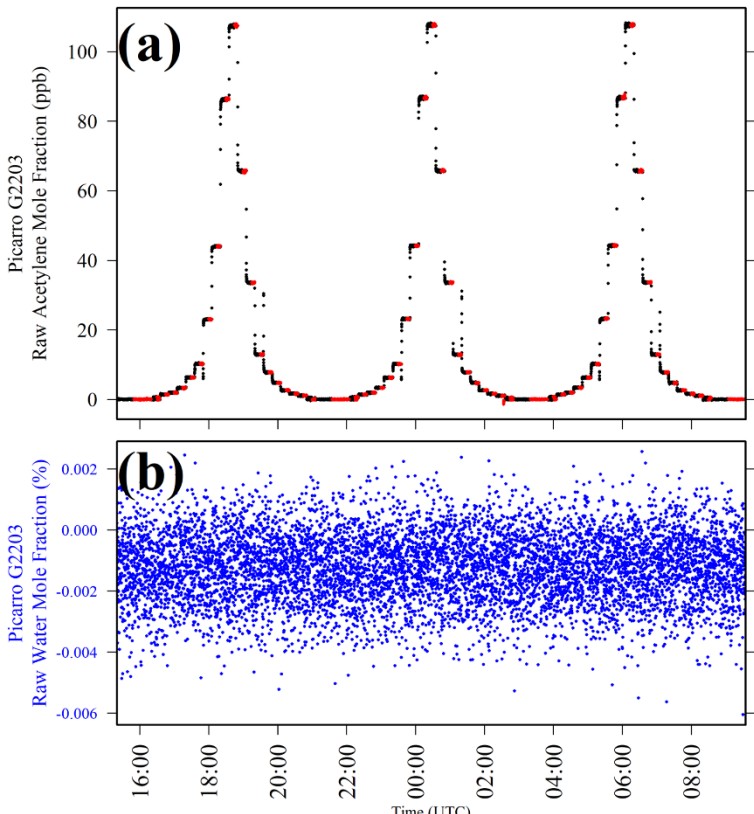

**Figure 6: (a) Picarro G2203 [C$_2$H$_2$]$_r$ plotted as black dots and (b) Picarro G2203 [H$_2$O]$_r$ plotted as blue dots, when sampling twenty different standard [C$_2$H$_2$] settings over three testing cycles by blending gas from the acetylene calibration cylinder with natural ambient compressed air. Periods used to derive averages are highlighted as red dots in (a).**

A linear regression was applied by comparing [C$_2$H$_2$] to [C$_2$H$_2$]$_r$ (presented in **Fig. 7**) for all [C$_2$H$_2$] settings except the lowest four, yielding a gain factor of 0.943 and an offset of −0.147 ppb, with a RMSE of ±0.0676 ppb. This calibration is only valid when sampling above the lowest stable [C$_2$H$_2$] level of 1.16 ppb (corresponding to [C$_2$H$_2$]$_r$ measurements of greater than 1.38 ppb). It may be possible to sample at a slightly lower [C$_2$H$_2$] level, but further exhaustive testing through trial and error would be required to precisely identify this threshold. The **Fig. 7** fit shows that when sampling at a fixed [C$_2$H$_2$] of 10 ppb, the Picarro G2203 reports 10.76 ppb [C$_2$H$_2$]$_r$. This +8% error is the same order of magnitude to the +12% error reported by Omara *et al.* (2016) for Picarro G2203 [C$_2$H$_2$]$_r$ measurements when sampling a 100 ppm [C$_2$H$_2$] standard. The error presented here could be significant when deriving tracer-based fluxes. This therefore emphasises the importance calibrating all [C$_2$H$_2$]$_r$ measurements obtained during field sampling. Although a calibration cannot be derived between 0 ppb and 1.16 ppb [C$_2$H$_2$] using this testing data, it can be concluded that sampling gas containing 0 ppb [C$_2$H$_2$] corresponds to a [C$_2$H$_2$]$_r$ measurement of 0.0125 ppb. This value corresponds to the average of the four measured 30-minute [C$_2$H$_2$]$_r$ averages obtained when sampling 0 ppb [C$_2$H$_2$] during the calibration test. It is also interesting that this value is different to the linear model zero intercept (or offset), which suggests that the Picarro G2203 behaves slightly differently in the absence of acetylene. Although this can be





used to correct $[C_2H_2]_r$ measurements when sampling air containing 0 ppb $[C_2H_2]$, it is not always possible to know if a $[C_2H_2]_r$ measurement made at this 0.0125 ppb level actually corresponds to sampling 0 ppb $[C_2H_2]$ or whether this erroneously corresponds to a slightly higher $[C_2H_2]$ level, which remains a limitation of using the Picarro G2203 tested in this work.

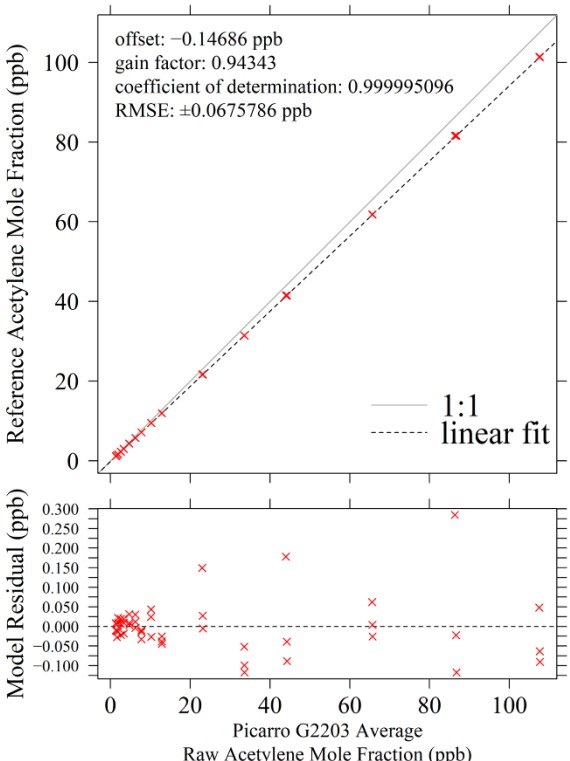

**Figure 7: (top) Picarro G2203 5-minute average $[C_2H_2]_r$ measurements, when combining gas from the acetylene calibration cylinder**
**with natural ambient compressed air, plotted against reference $[C_2H_2]$ levels (red crosses), with a linear regression model shown as a dashed black line and an identity line shown as a solid light grey line. (bottom) Corresponding model residuals between $[C_2H_2]_r$ and $[C_2H_2]$ (red crosses), with a 0 ppb $[C_2H_2]$ residual shown as a horizonal dashed black line.**

As an additional test, the calibration procedure was repeated but instead, using gas from the zero-air generator (UHP-300ZA-S, Parker Hannifin Manufacturing Limited, Gateshead, Tyne and Wear, UK) for dilution. Details of this test and presented in
**Sect. S5** in the **Supplement**. This test serves to check the validity of the acetylene calibration coefficients given above in a different gas mixture with no background levels of methane present. This additional test yielded a gain factor of 0.941 and an offset of +0.0014 ppb, with a RMSE of ±0.0356 ppb, when comparing standard $[C_2H_2]$ to $[C_2H_2]_r$. This gain factor is almost the same as when using natural ambient compressed air for dilution, with a similar offset close to zero. Nevertheless, small changes in $[C_2H_2]_r$ response in different background gases may have an influence on applications in field sampling, which
should be considered, although it does not appear to be so important for the Picarro G2203 tested here.



## 2.5 Measurement stability

As a final test, the stability of Picarro G2203 acetylene measurements was assessed by conducting an Allan variance ($\sigma_A{}^2$) test, which characterises the variability between sets of measurements over different timescales, ranging from the interval between consecutive measurements up until half of the duration of the test (although timescales of greater than a few hours hold little statistical value). First, natural ambient compressed air from a cylinder containing 2.076 ppm [CH$_4$] was blended with gas from the acetylene calibration cylinder to sample 10.9 ppb [C$_2$H$_2$]$_r$ (corresponding to 10.1 ppb [C$_2$H$_2$]), which was sampled for 12 hours. This blending assumes the MFCs to provide a constant flow rate, as any potential variability in MFC flow rate may be convolved with measurement noise, which is a limitation of this approach. Next, pure natural ambient compressed air from the same gas cylinder was sampled for 12 hours, corresponding to 0 ppb [C$_2$H$_2$]. To conduct this test, a similar set-up was used as for the acetylene calibration, as illustrated schematically in **Fig. 4**, when sampling 10.1 ppb [C$_2$H$_2$]. A modified version was used for the 0 ppb [C$_2$H$_2$] test to save gas, as no dilution is required; the compressed air cylinder was connected directly to a MFC, before subsequent humidification and drying.

To evaluate measurement stability, [C$_2$H$_2$]$_r$ measurements were first calibrated using calibration coefficients from the previous subsection when sampling 10.1 ppb [C$_2$H$_2$]. [C$_2$H$_2$]$_r$ measurements made at 0 ppb [C$_2$H$_2$] were corrected by subtracting 0.0125 ppb, as the linear model fit is not valid below 1.16 ppb [C$_2$H$_2$] (although, in theory, this offset correction has no effect in this analysis on variance). Then an $\sigma_A{}^2$ test was conducted on subsets of each prolonged dataset, as described by Werle *et al.* (1993). As the measurement frequency is inconsistent, the integration time was derived by finding the average of differences between the time corresponding to the first measurement in each subset and the first measurement in the next subset. In addition, the $\sigma_A{}^2$ test was repeated ten times by moving the starting and ending datapoint for each of the ten analyses, as the duration between each measurement follows a cycle of ten [C$_2$H$_2$]$_r$ measurements (as discussed above). These repeated tests were therefore used to obtain an average of the $\sigma_A{}^2$ values and corresponding integration times from the ten analyses.

Logarithmic plots showing Allan deviation precision ($\sigma_A$) as a function of integration time are given in **Fig. 8**, with white noise lines also shown. The $\sigma_A$ at the smallest integration time is $\pm 0.0863$ ppb (4.22 s integration time) when sampling 10.1 ppb [C$_2$H$_2$] and $\pm 0.0577$ ppb (4.24 s integration time) when sampling 0 ppb [C$_2$H$_2$]. Values of $\sigma_A$ at the smallest integration time did not use averaging of multiple measurements and simply took the variance between individual consecutive measurements, as each averaging bin contained one single element. **Figure 8** plots show consistently decreasing $\sigma_A$ with integration time, as expected, with a trend close to the white noise line. However, the [C$_2$H$_2$] plot at 0 ppm [C$_2$H$_2$] has an interesting feature in the first 100 s where $\sigma_A$ increases slightly before continuing its decline as a function of integration time, following a trend close to the white noise line (although offset from $\sigma_A$ at the lowest integration time). This could be associated with the roughly 40 s [C$_2$H$_2$]$_r$ measurement cycle, with the Picarro G2203 consistently struggling to fit for an absence of acetylene over each of the ten samples. The difference between the two **Fig. 8** fits supports the suggestion from the previous subsection, that the Picarro G2203 [C$_2$H$_2$] response is different at very low [C$_2$H$_2$] levels, close to 0 ppb. Nevertheless, neither $\sigma_A{}^2$ test shows a clear



sustained inflection in $\sigma_A$ decrease with integration time. This suggests that there is minimal drift over a 12-hour period compared to variability between individual measurements. This 12-hour duration is far shorter than a typical field sampling campaign (a few hours), demonstrating that Picarro G2203 measurements are unlikely to drift during field sampling. Furthermore, both tests reveal an $\sigma_A$ of less than ±0.1 ppb at the lowest integration time which suggests that variability between individual consecutive measurements is small, when sampling a single gas.

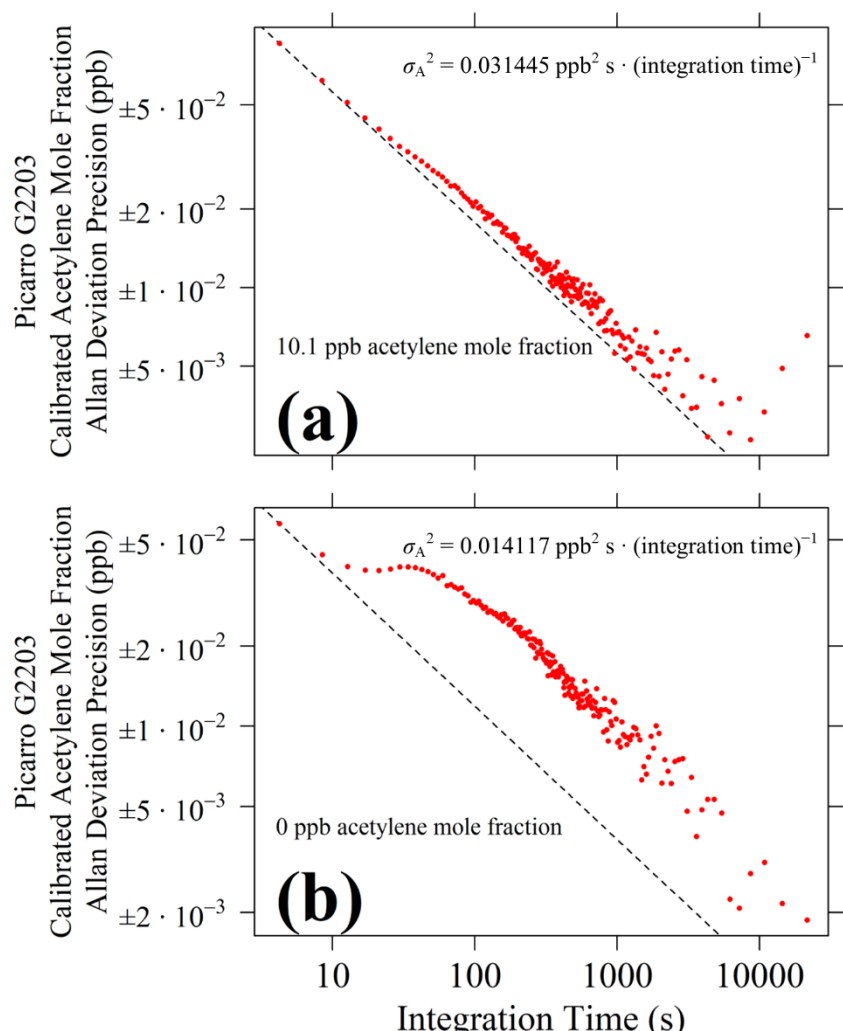

**Figure 8:** $\sigma_A$ **for Picarro G2203 calibrated [C$_2$H$_2$] measurements as a function of integration time derived from an average of ten different tests, plotted as red dots, when (a) sampling 10.1 ppb [C$_2$H$_2$] and (b) sampling 0 ppb [C$_2$H$_2$]. Logarithmic axes are used in each plot. Black dashed lines depict white noise, with each fit forced to intersect with $\sigma_A$ at the lowest integration time (the fitting coefficient is provided inside each plot).**



## 3 Field testing methane flux inversion method using an acetylene tracer release

### 3.1 Acetylene release method

Standard details on the acetylene release method itself are provided here. Acetylene is released from an 8.7 kg acetylene gas cylinder (Acétylène Industriel X50S, Air Products S.A.S., Saint Quentin Fallavier, France), with a 99.5% purity; as gas from such cylinders is naturally dry, it is released pure without any treatment. The cylinder is connected to an acetylene regulator (0783640, GCE Ltd, Warrington, UK), with a maximum output pressure of 1.5 bar. Gas flow is manually adjusted using a downstream metering valve (SS-4L, Swagelok Company). As the gauge pressure of gaseous acetylene must be kept below

1.5 bar for safety reasons, the pressure range of the acetylene regulator is targeted to between 0.5 bar and 1.0 bar whilst simultaneously adjusting the metering valve for the desired flow rate. As only 10% of the contents of the cylinder can safely be emitted per hour, the maximum sustained acetylene flow rate ($Q_{\text{acetylene}}$) level from this acetylene cylinder is 0.242 g s$^{-1}$. $Q_{\text{acetylene}}$ is measured using an acetylene flow meter (8C3B04-20X1/0, Cubemass C 300, Endress+Hauser Group Services AG, Reinach, Switzerland), which measures $Q_{\text{acetylene}}$ using the Coriolis technique (Baker, 2016) with an accuracy of no greater

than 0.00389 g s$^{-1}$ when sampling below 0.778 g s$^{-1}$ and no greater than 0.005 multiplied by $Q_{\text{acetylene}}$ itself when measuring above 0.778 g s$^{-1}$. Further details on the acetylene flow meter are provided in **Sect. S6** of the **Supplement**. Following each acetylene release, all equipment downstream of the regulator is flushed with nitrogen gas.

The acetylene release point is connected to the rest of the acetylene release equipment using Synflex 1300 tubing with an OD of 0.5 inches. A 0.5 inches to 0.25 inches standard SS Swagelok fitting reducer (SS-810-R-4, Swagelok Company) is used to

465 connect to this wider tube, which is chosen to minimise the pressure drop up to the release point. At the point of release, the tubing is split into four upwards-facing co-located Synflex 1300 tubes all with an OD of 0.5 inches, as illustrated in **Fig. 9**, to promote more even dissipation of the plume above ground level. A safety exclusion zone is designated as being 6 m away from the release point, with all acetylene release equipment placed outside this zone. This distance was calculated such that [C$_2$H$_2$] cannot exceed its lower atmospheric flammable limit of 2.5% (Williams and Smith, 1969) considering a $Q_{\text{acetylene}}$ of

470 0.5 g s$^{-1}$ and using Gaussian plume modelling in the worst-case Pasquill stability class (Turner, 1994), with a mean wind speed of 0.1 m s$^{-1}$ to accentuate [C$_2$H$_2$]. The boundaries of the safety exclusion zone are designated by cones, with the zone being constantly manned by the release operator who stands outside the zone and continuously surveys the acetylene release equipment.



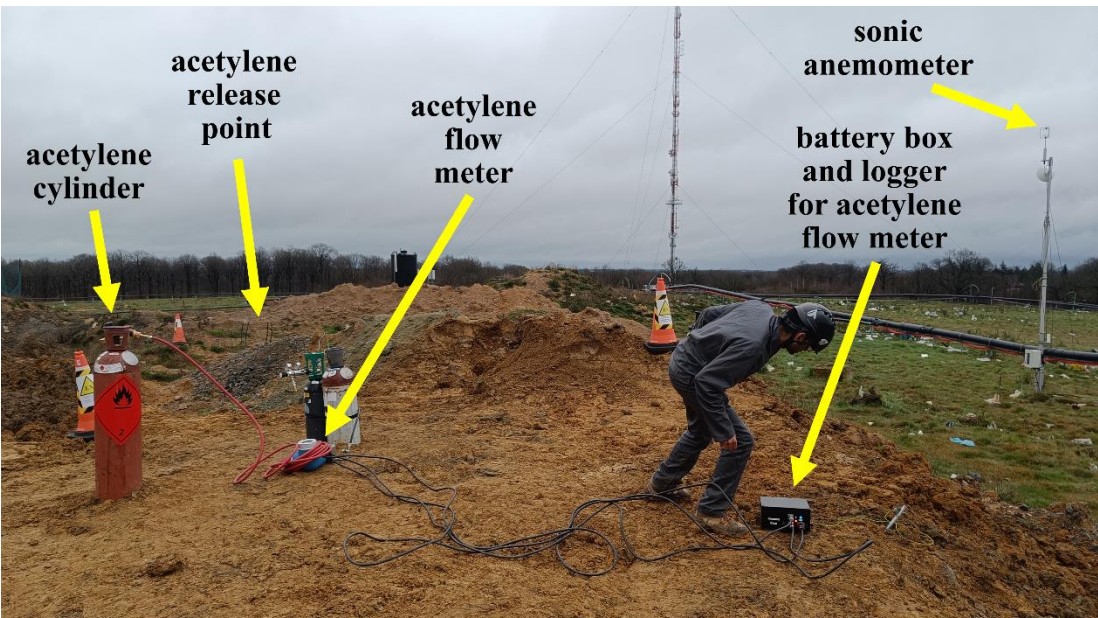

**Figure 9: A photograph of the acetylene release equipment when deployed during the campaign at the landfill site. The cones indicate the boundaries of the safety exclusion zone (not all cones are visible).**

Acetylene release equipment is protected from exposure to flammable gas mixtures using non-return valves. Flashback arrestors prevent an accidental flame from reaching upstream components and eventually, potentially entering the acetylene cylinder. A 5 bar flashback arrestor with a built-in non-return valve (50951, GCE Ltd) is connected directly downstream of the acetylene regulator. A 10 m reinforced high tensile synthetic textile acetylene hose (GCE Ltd) with an in-built non-return valve is used to connect the cylinder to the acetylene flow meter. An additional 2.0 bar flashback arrestor with a built-in non-return valve (Flashback Arrestor Super 66, WITT-Gasetechnik GmbH & Co KG, Witten, Germany) is connected upstream of the acetylene flow meter. The entire acetylene release set-up is illustrated schematically in **Fig. 10**. All fittings and connectors were selected for compatibility with acetylene, with any copper alloys (including brass) containing no more than 65% copper.





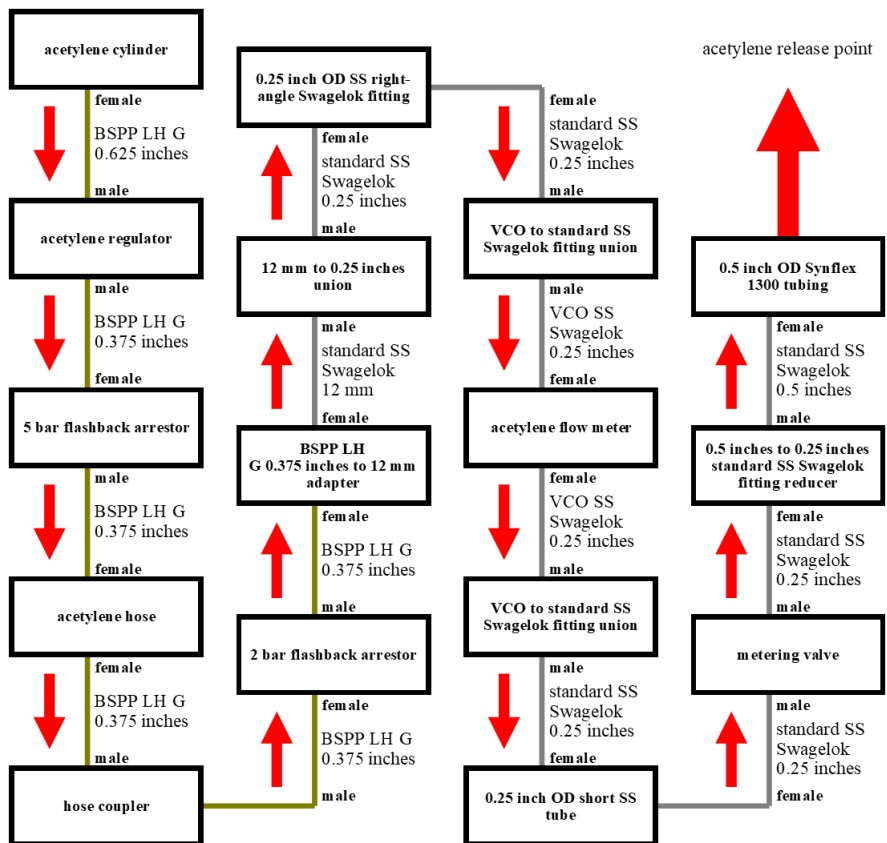

**Figure 10: A schematic of each individual component (in black boxes) used when conducting an acetylene release. Dark yellow lines indicate brass connections and grey lines indicate SS connections. The thread type between each component is given next to each line and the gender of the threads is given in bold text outside of each box. The direction of acetylene gas flow is indicated by red arrows.**

Equipment for use with acetylene conventionally uses threads for British Standard Pipe parallel (BSPP) left-hand (LH) G fittings, as opposed to right-hand fittings which are conventionally used in most other applications. These BSPP LH G fittings are converted into threads for standard SS Swagelok fittings using a SS 12 mm to 0.25 inches union (SS-12M0-6-4, Swagelok Company) and a brass BSPP LH G 0.375 inches to 12 mm adapter. In addition, the inlet and outlet for the acetylene flow meter has threads for VCO SS Swagelok fittings. A VCO to standard SS Swagelok fitting union (SS-4-VCO-6-400, Swagelok

Company) is therefore used to obtain threads for standard SS Swagelok fittings, for integration with the rest of the set-up.

**3.2 Landfill site release campaign**

To test our acetylene release and corresponding $Q_{\text{methane}}$ calculation methods, an acetylene release was conducted from within a landfill site. This particular landfill site was chosen as it a known facility-scale methane source, for which we were able to acquire site access. The specificities of this specific study site (for example waste content, waste quantity, site age and site

management) are irrelevant in this study; this study is dedicated to the acetylene release method itself in the context of $Q_{\text{methane}}$




quantification methods in general. Therefore, the magnitude of any derived $Q_{methane}$ rate is beyond the scope of this study and will be discussed in a future publication. An aerial photograph of the site is shown in **Fig. 11**. The acetylene release location from within the site was selected due to accessibility with regards to transportation and installation of a heavy acetylene cylinder. We were not authorised to conduct a release from a more central location due to site activities and the presence of

505 active open landfill cells. In general, the acetylene release location should be as close to the source as possible, to trace emission of the methane source as it disperses through the atmosphere, although this can be difficult from a complex heterogeneous source such as a landfill site (Fredenslund *et al.*, 2019). The complications associated with tracer release positioning is discussed in further detail in **Sect. 4**. A three-dimensional sonic anemometer (WindMaster Pro, Gill Instruments Limited, Lymington, Hampshire, UK) measured winds at 20 Hz, near to the acetylene release point, as illustrated in **Fig. 10**, which was

510 visually aligned with an uncertainty of ±4°.

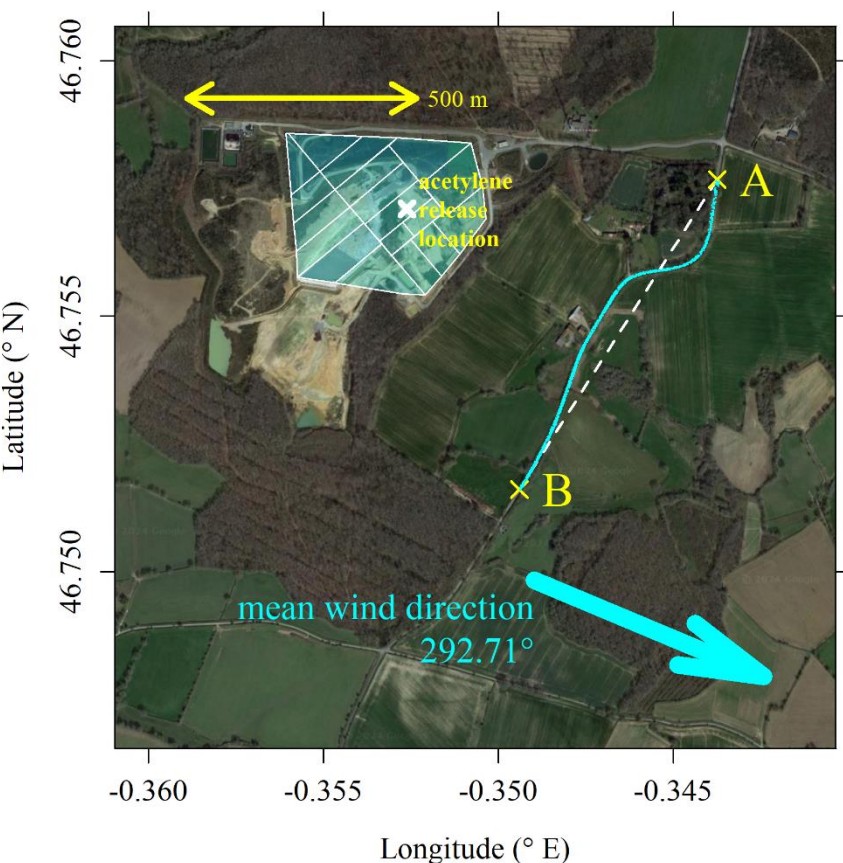

**Figure 11: The location of the acetylene release (white cross) which was conducted at ground level, plotted on top of a background map. Transparent shaded cyan polygons indicate both active and inactive cells, as identified by the landfill site operator. The location of Picarro G2203 [C$_2$H$_2$] measurements are shown as cyan dots on a sampling road between Point A and Point B, indicated by the yellow crosses. The plane between Point A and Point B is shown as a dashed white line. The average direction in which the wind vector was blowing is shown as a cyan arrow, as recorded by the sonic anemometer, from 20 minutes before the first transect until the end of the final transect. The background image is taken from © Google Maps (imagery (2024): Maxar Technologies).**



Rather than relying on stationary (zero dimensional) downwind sampling, the methane and acetylene plumes were instead sampled through multiple plume transects for subsequent integration, as $Q_{methane}$ derived from one-dimensional transects results in improved flux accuracy (as discussed in **Sect. 1**). Twenty vehicular transects were conducted on a nearly downwind sampling road during the acetylene release. The position and nature of downwind transects can have an impact on $Q_{methane}$ estimates if the acetylene release location is not perfectly co-located with the methane source, which is discussed in further detail in **Sect. 4**. The vehicle was equipped with the Picarro G2203 gas analyser, for which all sampled air passed through the Nafion-based gas dryer followed by a magnesium perchlorate scrubber. The air inlet was fixed to the roof of the vehicle. The Picarro G2203 was powered using a portable mains power supply bank. A LI-COR LI-7810 (LI-COR, Inc.) gas analyser was also installed in the vehicle which shared the same air inlet (but no dryer), measuring $[CH_4]_r$ and $[H_2O]_r$ at a frequency of approximately 1 Hz. The LI-COR LI-7810 was powered by its internal battery. A global navigation satellite system (GNSS) positional logger made measurements of vehicular position at 1 Hz. The timestamp of Picarro G2203 and LI-COR LI-7810 measurements were individually adjusted to GNSS time by breathing into the air inlets at a fixed GNSS time and recording the time of the $[CH_4]_r$ responses. This could be achieved as a member of our team exhales methane; most humans do not exhale detectable methane enhancements (Dawson *et al.*, 2023).

The sampling campaign duration is defined as 20 minutes before the start of first transect up to the time of the point of the final transect. $Q_{acetylene}$ was largely stable for the full duration of the sampling campaign, with an average $Q_{acetylene}$ level of $0.239 \text{ g s}^{-1}$ and a standard deviation variability of $\pm0.001 \text{ g s}^{-1}$, as presented in **Sect. S7** in the **Supplement**. The 20-minute period of continuous acetylene flow in advance of vehicular sampling allowed the acetylene plume to become well established and to stabilise in the atmosphere (Fredenslund *et al.*, 2019). The average wind direction was 292.7° with respect to True North and the average wind speed was $3.84 \text{ m s}^{-1}$, for the duration of the sampling campaign (see **Sect. S7** in the **Supplement**).

The limits of the sampling road for $Q_{methane}$ calculation purposes were defined as being between Point A and Point B (indicated in **Fig. 11** as yellow crosses); although vehicular sampling protruded these points to sample on a longer stretch of road, the landfill emission plumes remained within this spatial range, serving as sensible limits for subsequent analysis. All Picarro G2203 $[C_2H_2]_r$ measurements from each transect are projected onto the vertical plane between Point A and Point B in **Fig. 12** (a). During six transects (transect 2, 5, 9, 14, 17 and 18), a feature containing $[C_2H_2]_r$ measurements of less than $-0.5$ ppb was observed. These negative $[C_2H_2]_r$ measurements were observed just before observing the acetylene peak, making these periods clearly distinguishable from instrumental noise. Furthermore, consistent negative $[C_2H_2]_r$ measurements were made during each feature, as opposed to noise which generally varied between randomly positive and negative measurements. These erroneous measurements were probably due to complications associated with the Picarro G2203 internal spectral fitting algorithms, in response to a sudden sharp $[C_2H_2]$ change. This may have been due to mis-fitting issues as the Picarro G2203 takes some time to complete a scan against all wavelengths across the acetylene IR absorption peak. These six transects have therefore been removed from the subsequent flux analysis, resulting in fourteen remaining successful transects.





**Figure 12: (a)** Raw Picarro G2203 $[C_2H_2]_r$ measurements, **(b)** calibrated Picarro G2203 $[C_2H_2]_r$ measurements, **(c)** calibrated Picarro G2203 $[CH_4]$ measurements, **(d)** calibrated LI-COR LI-7810 $[CH_4]$ measurements, **(e)** calibrated LI-COR LI-7810 $[CH_4]$ measurements interpolated to the Picarro G2203 $[C_2H_2]_r$ timestamp and **(f)** calibrated LI-COR LI-7810 $[CH_4]$ measurements interpolated to the Picarro G2203 $[C_2H_2]_r$ timestamp with $[CH_4]$ below $[CH_4]_{threshold}$ set to $[CH_4]_0$ for each transect, all plotted as coloured dots (see legend for transect colours) on the plane between Point A and Point B downwind of a landfill site. Only successful transects are shown in **(b)** and **(f)**.

For the fourteen remaining transects, all $[C_2H_2]_r$ measurements above 1.38 ppb from the Picarro G2203 were converted into dry calibrated measurements using laboratory-derived coefficients from **Sect. 2**. These calibrated $[C_2H_2]$ measurements are presented in **Fig. 12 (b)**, as a function of distance along the plane between Point A and Point B. All Picarro G2203 $[C_2H_2]_r$ measurements of less than 1.38 ppb were fixed to 0 ppb $[C_2H_2]$, as this sampling may be unstable and some non-zero $[C_2H_2]$ sampling in this range can erroneously resolve to the $[C_2H_2]_r$ observed at 0 ppb $[C_2H_2]$ (as discussed in **Sect. 2**). The influence of this step when applied to low (but non-zero) $[C_2H_2]$ sampling, on $Q_{methane}$, is dealt with in the next subsection. Picarro G2203 $[CH_4]_r$ measurements from all twenty transects were converted in dry calibrated $[CH_4]$ using the coefficients provided in **Sect.**



**S1** in the **Supplement**. All $[CH_4]_r$ measurements from the LI-COR LI-7810 were converted into dry calibrated $[CH_4]$, by first
applying an empirical water correction followed by a calibration which could be cross-referenced to standards on the WMO
greenhouse gas scale for methane (WMO X2004A). Calibrated Picarro G2203 and LI-COR LI-7810 $[CH_4]$ measurements
from each transect are also shown in **Fig. 12** (c and d, respectively), as a function of distance along the plane between Point A
and Point B.

### 3.3 Landfill site methane emission flux calculation

In this study, two sets of $Q_{methane}$ values were calculated using both $[C_2H_2]$ as well as $[C_2H_2]_r$ as model input, to compare any
influence of Picarro G2203 acetylene calibration on $Q_{methane}$ results. The principle of using acetylene as a tracer gas for methane,
requires mole fraction measurements from both the acetylene and the methane plume to calculate $Q_{methane}$. $Q_{methane}$ was derived
in this work by integrating the observed methane and acetylene emission plume as a function of distance along the sampling
road. This form of spatial integration is used to apply equal weighting to each mole fraction measurement as a function of
distanced covered, especially with irregular spatial measurements due to irregular driving speed and sampling frequency.
Although, in theory, $Q_{methane}$ can be derived from a single downwind measurement point (as discussed in **Sect. 1**), spatial
integration results in better accuracy, especially if the methane and acetylene plumes do not perfectly overlap, as illustrated in
**Fig. 12**.

Before conducting this integration, LI-COR LI-7810 $[CH_4]$ measurements were first interpolated to the lower frequency (and
less regular) Picarro G2203 $[C_2H_2]$ timestamp so that each $[C_2H_2]$ had a corresponding spatial $[CH_4]$ measurement, as shown
in **Fig. 12** (e). In general, the likelihood of sampling close to the maximum of each emission plume decreases with larger
sampling gaps. Yet, due to the far superior LI-COR LI-7810 $[CH_4]$ sampling frequency, this data was assumed to capture the
full methane plume shape, allowing the loss of sampling points from this plume to replicate the data loss from the acetylene
plume. Due to the intermediate Picarro G2203 $[CH_4]$ sampling frequency, these measurements were not used in this analysis.
By contrast, interpolating Picarro G2203 $[C_2H_2]$ to the LI-COR LI-7810 $[CH_4]$ would be less appropriate as acetylene plume
measurements would be artificially generated from a lack of information on acetylene plume shape (*i.e.* artificial gap filling).
As a general caveat, perfect replication of methane plume information loss requires the methane and acetylene plumes to
perfectly overlap in space. As the plumes were slightly offset (see **Fig. 12**), this interpolation method did not result in identical
information loss from both the acetylene plume and the methane plume (see **Sect. 4** for discussion). Nevertheless, interpolation
to the lower Picarro G2203 $[C_2H_2]$ timestamp ensured that the likelihood of loss of information on the methane emission plume
using interpolated LI-COR LI-7810 $[CH_4]$ measurements remained the same, thereby avoiding any bias in multiple $Q_{methane}$
estimates derived from individual transects.

$Q_{methane}$ also requires background ($[C_2H_2]_0$ and $[CH_4]_0$) values to characterise the enhancement of the tracer and methane
emission plume above the background. $[CH_4]_0$ was derived by taking the average of the lowest five non-interpolated $[CH_4]$



measurements from each transect. This accounts for $[CH_4]_0$ natural regional variability over time. This also corrects for any $[CH_4]$ measurement offset that may occur due to instrumental drift. For acetylene, $[C_2H_2]_0$ was fixed to 0 ppb for all transects, as negligible levels of acetylene are expected in the natural ambient background. Thus, this approach does not account for potential changes in $[C_2H_2]_r$ measurement offset. While taking the lowest five $[C_2H_2]_r$ measurements from each transect was considered for an uncalibrated $[C_2H_2]_0$, due to the noisy baseline, this would inevitably result in capturing the noise's weakest

values, making this method unsuitable. For the calibrated $[C_2H_2]$ data (derived following the procedure outline above), sensor baseline drift cannot be deduced from the lowest measurements, as all $[C_2H_2]_r$ of less than 1.38 ppb are fixed to 0 ppb $[C_2H_2]$. But with the stability of CRDS (Crosson, 2008, Yver-Kwok *et al.*, 2021), drift is unlikely to be a major issue, which is supported by the $\sigma_A^2$ test results given in **Sect. 2** which showed no $[C_2H_2]$ drift over a period of 12 hours.

As an additional step it is important to take into account the potential loss of low (but non-zero) $[C_2H_2]$ sampling due to setting

a maximum $[C_2H_2]_r$ threshold of 1.38 ppb (corresponding to 1.16 ppb $[C_2H_2]$), below which all $[C_2H_2]$ is fixed at 0 ppb. In theory, this is not likely to be an issue away from of the acetylene plume, as there are no major acetylene sources and $[C_2H_2]_0$ is expected to consistently equal 0 ppb. However, a small number of non-zero $[C_2H_2]$ enhancements may be lost from the edges of each plume. To replicate this effect on $[CH_4]$ measurements, any interpolated $[CH_4]$ measurements below a methane mole fraction threshold ($[CH_4]_{threshold}$) from each of the fourteen successful transects were fixed to $[CH_4]_0$, as illustrated in **Fig.**

**12** (f). $[CH_4]_{threshold}$ is calculated for each successful transect using

$$[CH_4]_{threshold} = \left( \left( \frac{\text{maximum } [CH_4] - [CH_4]_0}{\text{maximum } [C_2H_2] - [C_2H_2]_0} \right) \cdot 1.16 \text{ ppb} \right) + [CH_4]_0, \tag{3}$$

which takes the ratio between maximum mole fraction enhancements from each transect. Although this is not a perfect approach as the maximum height of both the methane and acetylene plume were unlikely to be captured (due to large sampling gaps and an offset acetylene plume), the distance of the maximum $[C_2H_2]$ and $[CH_4]$ measurements from the acetylene and

methane plume centres, respectively, should average out over a sufficient number of transects, resulting on a null overall effect on $Q_{methane}$. These modified $[CH_4]$ values must be used alongside calibrated $[C_2H_2]$ measurements, during $Q_{methane}$ calculation. However, $Q_{methane}$ derived using uncalibrated $[C_2H_2]_r$ does not require modified $[CH_4]$, as this tests flux estimation assuming all uncalibrated $[C_2H_2]_r$ measurements to be correct. **Fig. 12** (e and f) shows that interpolated $[CH_4]$ measurements without this threshold are similar to modified $[CH_4]$ measurements with the imposed threshold. Nevertheless, this step is important to

minimise the effect of inflated methane plumes due to erroneously low $[C_2H_2]$ measurements (without corresponding erroneously low $[CH_4]$ measurements) from biasing $Q_{methane}$.

$Q_{methane}$ was calculated following

$$Q_{methane} = Q_{acetylene} \cdot \left( \frac{\sum_{i=2}^{n-1} \left( ([CH_4]_i - [CH_4]_0) \cdot \Delta x_i \right)}{\sum_{i=2}^{n-1} \left( ([C_2H_2]_i - [C_2H_2]_0) \cdot \Delta x_i \right)} \right) \cdot \left( \frac{M_{methane}}{M_{acetylene}} \right), \tag{4}$$



where $i$ represents each individual measurement within each transect and $n$ represents the total number of measurements within each transect. $\Delta x$ is the average spatial distance between adjacent measurements given by

$$\Delta x_i = \frac{\Delta x_{i-1 \to i} + \Delta x_{i \to i+1}}{2},\tag{5}$$

where $\Delta x_{a \to b}$ is the spatial distance between any measurement point $a$ and any other measurement point $b$. $\Delta x_{a \to b}$ is derived using the difference in latitude and longitude between point $a$ and point $b$. Equation (4) requires $[CH_4]$ and $[C_2H_2]$ to be in the same mole fraction units (*e.g.* ppm) and for both mole fractions to be either dry or wet (dry mole fractions are used here). $M_{methane}$ is the molar mass of methane ($16.0425 \text{ g mol}^{-1}$) and $M_{acetylene}$ is the molar mass of acetylene ($26.0373 \text{ g mol}^{-1}$).

## 4 Results and discussion

### 4.1 Landfill flux results

Landfill $Q_{methane}$ results derived using Eq. (4) are presented in **Fig. 13**. The $Q_{methane}$ average from the fourteen individual vehicular transects is $17.3 \text{ g s}^{-1}$, with a standard deviation variability between the different transect fluxes of $\pm 9.6 \text{ g s}^{-1}$, when using calibrated $[C_2H_2]$ (and $[CH_4]$ fixed to $[CH_4]_0$ below $[CH_4]_{threshold}$). The significance of $Q_{methane}$ in the context of overall landfill emissions from this specific study site and in comparison to $Q_{methane}$ derived using other methods will be discussed in a forthcoming study. A combined $Q_{methane}$ was also derived by combining data from all successful transects simultaneously within Eq. (4) (where separate $[C_2H_2]_0$ and $[CH_4]_0$ values were subtracted from data corresponding to each transect) to yield a single combined emission flux estimate of $15.8 \text{ g s}^{-1}$. This is consistent with the average of the fourteen individual fluxes, within the standard deviation uncertainty range. Using the uncalibrated $[C_2H_2]_r$ data as Eq. (4) input (and unaltered interpolated $[CH_4]$) yielded smaller $Q_{methane}$ values, with an average of the fourteen individual vehicular transects of $16.0 \text{ g s}^{-1}$ and a standard deviation variability between the different transects of $\pm 8.6 \text{ g s}^{-1}$. This represents a flux underestimation of $-7.6\%$ compared to $Q_{methane}$ derived using calibrated $[C_2H_2]$ as Eq. (4) input.



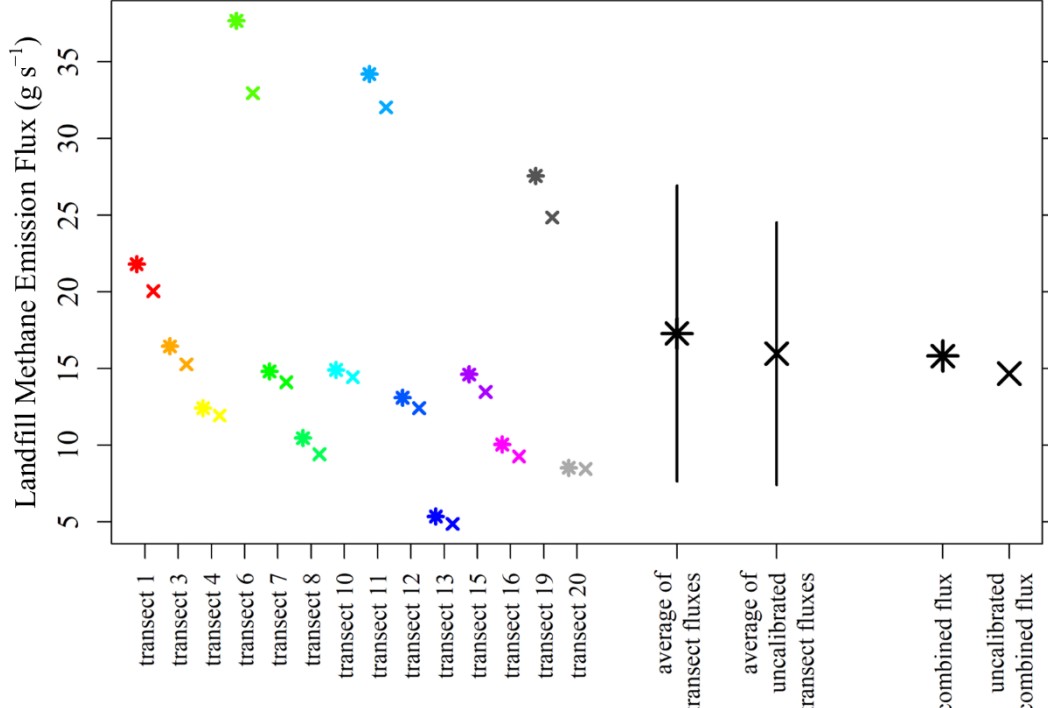

**Figure 13: Individual landfill methane flux estimates for each individual transect, corresponding averages and the overall combined methane emission flux estimate, plotted as stars using calibrated [C$_2$H$_2$] (with [CH$_4$] below [CH$_4$]$_{threshold}$ fixed to [CH$_4$]$_0$) and crosses using uncalibrated [C$_2$H$_2$]$_r$ as Eq. (4) input. The black lines indicate the standard deviation variability between individual methane transect fluxes.**

**4.2 Discussion**

**Figure 13** shows that there is a large disparity between $Q_{methane}$ estimates for individual transects, with a 56% standard deviation variability; which is primarily due to differing turbulent patterns between methane and acetylene plume dispersion, partially caused by a suboptimal acetylene release location (discussed below). This emphasises the importance of conducting a sufficient number of transects to average over this variability, representative of the true emission flux. Although an aspect of calculated $Q_{methane}$ variability may be due to variability in true landfill methane emissions (which is a limitation of this work), this is

expected to be relatively small over the limited sampling window (less than 3 hours). Landfill emissions are relatively consistent in the absence of abrupt environmental or operational changes. Thus, a constant true methane emission flux value is assumed for all transects, with observed $Q_{methane}$ variability between transects driven by limitations in sampling and the nature of the tracer release (discussed below). $Q_{methane}$ for transect 13 was particularly low; **Fig. 12** (f) shows that this was due to a disproportionately narrow methane plume. If the methane and acetylene plume were to share better spatial overlap, this

issue would likely diminish, as both methane and acetylene plumes would be equally small at the time and location of measurement. Transect 20 is similarly results in low $Q_{methane}$, where a small methane plume was detected. Conversely, transect 13 resulted in a small $Q_{methane}$ due to a large acetylene plume. Transect 6 resulted in the largest $Q_{methane}$ due to a large methane



plume. However, transect 7, which included the largest [$CH_4$] measurement (of the fourteen successful transects) did not result in such a high $Q_{methane}$ due to accompanied detection of a substantially sized acetylene plume.

Previous studies have shown that tracer poor localisation with the methane emission source can cause $Q_{methane}$ variability (Mønster *et al.*, 2014, Yver Kwok *et al.*, 2015, Ars *et al.*, 2017), as observed across the fourteen transects fluxes presented here (see **Fig. 13**), resulting in a poor methane and acetylene plume overlap (see **Fig. 12**). Good tracer and methane source co-location is essential for accurate tracer-based fluxes (Delre *et al.*, 2018, Fredenslund *et al.*, 2019, Lui *et al.*, 2024). This ensures good mixing of the entire tracer plume with the methane plume (Matacchiera *et al.*, 2019) with identical dispersion (Johnson
*et al.*, 1994, Lamb *et al.*, 1995, Daube *et al.*, 2019, Mønster *et al.*, 2019), such that even a mole fraction ratio at any single point yields an accurate emission flux (Omara *et al.*, 2016, Ars *et al.*, 2017). A large disperse methane emission facility such as a landfill site may require multiple acetylene release points for improved plume overlap, such that the individual acetylene plumes overlap into a larger overall acetylene plume more representative of the shape of the complex methane emission plume emanating from the heterogenous surface emission source (Scheutz and Fredenslund, 2019, Matacchiera *et al.*, 2019, Mønster
*et al.*, 2019, Vechi *et al.*, 2022). Yet, organising an acetylene release from other central locations at our landfill study site was challenging, especially from over active waste. Authorisation was secured many months in advance, making short term on-site changes difficult to implement. Furthermore, the ideal tracer release location can be unique to each site (Matacchiera *et al.*, 2019), making it difficult to anticipate. Tracer localisation issues can be addressed by using a hybrid approach such as that of Ars *et al.* (2017), which combined tracer-based fluxes with a statistical inversion and an atmospheric transport model, for
significantly improved overall flux estimates despite poor tracer localisation.

In conjunction with acetylene release location, good downwind positioning is essential for good plume mixing and overlap (Scheutz *et al.*, 2011, Daube *et al.*, 2019), to allow sufficient distance between the sources and the sampling location (Galle *et al.*, 2001, Feitz *et al.*, 2018), so the full extent of the emission plume is captured (Mønster *et al.*, 2015, Delre *et al.*, 2018). Tracer-based fluxes are fundamentally limited to locations with downwind site access (Bell *et al.*, 2017). Yet our study had
685 limited sampling options, with only one near-site downwind sampling road. For plumes that do not perfectly overlap (as in this study), integrating along the sampling road requires the road to be straight and perpendicular to wind direction (Yacovitch *et al.*, 2017), to avoid the methane and acetylene plumes being detected at different distances (Ars *et al.*, 2017). Measurements closer to the site have a higher mole fraction with respect to the plane perpendicular to wind direction. But if the sampling road is nearly straight and perpendicular to wind direction (assumed here), the importance of these errors declines. In addition,
sampling a sufficient distance from the source can reduce issues due to poor tracer co-location with the source, as the two plumes have more time to mix (Fredenslund *et al.*, 2019).

Although a perfectly co-located acetylene release yields ideal $Q_{methane}$ estimates, it is also possible to derive $Q_{methane}$ if the acetylene source is slightly offset from the methane source (Mønster *et al.*, 2014), as in this study. In such a scenario, the two plumes will be detected at different downwind locations (Ars *et al.*, 2017), but similar plume dispersion allows the ratio





between these two integrated plumes to yield a flux. Yet this requires the two sources to be an equal distance from the sampling road, perpendicular to wind direction (Mønster *et al.*, 2014, Ars *et al.*, 2017, Daube *et al.*, 2019). It also requires identical wind conditions for the duration of each transect. Otherwise, the methane and acetylene plumes may have dispersed under different conditions upon detection, resulting in lower mole fraction measurements during higher winds and *vice versa*. A similar amount of information is also required from each plume (Delre *et al.*, 2018), which was achieved in this work by interpolating

to the lower (Picaro G2203 [$C_2H_2$]) timestamp, thereby avoiding the contentious practice of gap filling. Although it is difficult to perfectly satisfy all above conditions, getting close enough can yield acceptable $Q_{methane}$ estimates.

Our [$CH_4$] interpolation approach ensured that each [$C_2H_2$] measurement had a corresponding spatiotemporal [$CH_4$] measurement. This allowed all measurements to be integrated as a function of distance along the sampling road, using the summation given by Eq. **(4)**. An alternative integration approach is to continuously model the emission plumes as a function

of distance, for analytical integration rather than summation (Fredenslund *et al.*, 2019), although this requires a sufficient sampling density to characterise plume shape (Delre *et al.*, 2018). This latter method would be challenging with Picarro G2203 instrument used in this work due to its irregular [$C_2H_2$]$_r$ sampling frequency, resulting in large gaps on plume dispersion information from downwind transects, with a maximum measurement time gap of 13 s. Interpolating the higher frequency LI-COR LI-7810 [$CH_4$] to the lower frequency Picarro G2203 [$C_2H_2$] timestamp mirrored this data loss in our summation

integration (Eq. **(4)**). The average measured (0.3±0.1) Hz [$C_2H_2$] sampling frequency of the Picarro G2203 is lower than 0.5 Hz proposed by the manufacturer (Picarro, Inc., 2015). This may be due to the age and irregular operation of this specific Picarro G2203 gas analyser, which was manufactured in September 2015. It is important to state that the Picarro G2203 used in this study experienced some spectral fitting issues in the past. These were resolved following manufacturer support. However, this may have inadvertently resulted in the irregular sampling frequency as a residual unresolved issue. During a previous testing

campaign, we experienced instrument failure during excessive acetylene exposure, which may explain this effect, although we cannot be certain. The slow sampling rate may also be associated with the unusual behaviour observed when sampling wet air, with [$H_2O$]$_r$ peaks and unpredictable [$C_2H_2$]$_r$ response, as described in **Sect. 2**.

The [$C_2H_2$]$_r$ response of the Picarro G2203 tested in this work was calibrated by diluting gas from an acetylene calibration cylinder. Methane served as a proxy gas to indicate the level of dilution achieved at each MFC blending step, by diluting gas

from a methane calibration cylinder in the exact same way. Thus, any potential MFC blending errors should cancel out in this approach. Yet, there is a ±3% [$C_2H_2$] uncertainty in the acetylene calibration cylinder, which is a limiting factor in this method. Fredenslund *et al.* (2019) propose that uncertainty in calibration mole fraction can be treated as a random uncertainty in overall flux estimates (given as 8% in their study). Perhaps such an uncertainty should instead be treated as a bias, as a poorly calibrated instrument will consistently bias flux estimates in one direction. Nevertheless, the cylinder mole fraction uncertainty is small

in this work, compared to the correction induced from the derived calibration which reduces a 10 ppb [$C_2H_2$]$_r$ measurement down to 9.29 ppb [$C_2H_2$].





Yet it is concerning that the Picarro G2203 reports unstable $[C_2H_2]_r$ measurements when sampling at low (but non-zero) $[C_2H_2]$ levels, with a maximum stable $[C_2H_2]$ level of 1.16 ppb observed during testing. In this low $[C_2H_2]$ range, Picarro G2203 $[C_2H_2]_r$ measurements occasionally resolved to the $[C_2H_2]_r$ level observed at 0 ppb $[C_2H_2]$. This means that any $[C_2H_2]_r$
measurement made at below 1.38 ppb bears some uncertainty, as it could correspond to any $[C_2H_2]$ level between 0 ppb and 1.16 ppb $[C_2H_2]$, although this upper uncertainty limit may be reduced slightly with further calibration testing. In this work, it was assumed that $[C_2H_2]_0$ in ambient air is 0 ppb. Therefore, measurements made away from an observed acetylene plume were assumed to equal 0 ppb $[C_2H_2]$, with the key uncertainty occurring at the edges of the observed plume, which were also fixed to 0 ppb $[C_2H_2]$ in this work. This was dealt with by additionally fixing some $[CH_4]$ measurements (below $[CH_4]_{threshold}$)
from each transect to $[CH_4]_0$, to avoid biasing $Q_{methane}$ results. Thus, a subset of both $[C_2H_2]$ and $[CH_4]$ measurements from each transect were forced to their corresponding background levels.

When deriving tracer-based fluxes, it is also important to evaluate the magnitude of peak mole fraction enhancements above instrumental noise, to ensure that the plume is detectable and can be characterised when subtracted from the background mole fraction level (Yver Kwok *et al.*, 2015, Ars *et al.*, 2017, Yacovitch *et al.*, 2017, Delre *et al.*, 2018, Fredenslund *et al.*, 2019).
The average $[C_2H_2]$ peak height of (15±3) ppb for the fourteen useable transects is far superior than the 0.24 Hz $\sigma_A$ of ±0.0863 ppb at 10.1 ppb $[C_2H_2]$. Thus, we can conclude that the acetylene release emission flux was sufficiently high and downwind sampling distance was sufficiently close to the source to detect acetylene emission plumes with a sufficiently high $[C_2H_2]$ resolution.

The $Q_{methane}$ results presented in this work emphasise the importance of calibrating all gas mole fraction measurements, even
those of a tracer gas. To our knowledge, a small number of previous studies using the Picarro G2203 have tested its $[C_2H_2]_r$ measurement response (Mønster *et al.*, 2014, Omara *et al.*, 2016), with no previous studies demonstrating a calibration with which to correct Picarro G2203 $[C_2H_2]_r$ measurements. In this study, using raw uncalibrated $[C_2H_2]_r$ measurements as Eq. **(4)** input resulted in $Q_{methane}$ estimates that were consistently lower than corresponding estimates using calibrated $[C_2H_2]$ measurements, with an average underestimation of 7.6%. This is a key outcome of this study. Various previous studies have
derived tracer-based fluxes downwind of controlled tracer releases, reporting various uncertainty ranges (Mønster *et al.*, 2014, Feitz *et al.*, 2018). For example, Ars *et al.* (2017) reported a tracer-based methane flux uncertainty of +14% for a controlled tracer release that was perfectly co-located with the methane emission source, although each tracer release study is unique can cannot be compared directly (for example, higher tracer release rates can result in a lower uncertainty). Liu *et al.* (2024) reported an average tracer-based error of 19% compared to known emission fluxes from thirteen integrated transects.
Fredenslund *et al.* (2019) conducted an uncertainty budget for their tracer-based flux approach, with an overall error of 15%. Yet the overall uncertainty from these previous studies is generally given as a random uncertainty and does not distinguish between systemic emission bias due to lack of tracer gas calibration and other random methodological errors. This bias may be an important contributory factor within the overall uncertainty of tracer-based flux estimates. For example, consistently



using $[C_2H_2]_r$ measurements from our Picarro G2203 would consistently underestimate Eq. (**4**) $Q_{methane}$; this is more concerning than random methodical uncertainties which, should cancel each other out over a sufficiently large sampling average. Such biases may be propagated which may influence our understanding of the importance of certain facility-scale emission sources contributing towards the global methane budget.

## 5. Conclusions

The first detailed characterisation and calibration of acetylene measurements made by the Picarro G2203 gas analyser is presented. Initially, water response was tested. This showed that characterising $[C_2H_2]_r$ response as a function of $[H_2O]$ is unfeasible, due to an inconsistent $[C_2H_2]_r$ response. Furthermore, $[H_2O]_r$ measurements appear to episodically erroneously spike at low $[H_2O]$ levels, making any such water characterisation of $[C_2H_2]_r$ response prohibitive. Instead, $[C_2H_2]_r$ was characterised in dry conditions. As there are no readily available acetylene gas standards, gas from an acetylene calibration cylinder (containing 10 180 ppb $[C_2H_2]$) was diluted. To quantify the level of dilution, methane was used as a proxy gas, by first diluting gas from a cylinder containing methane. The observed $[CH_4]$ level was compared to the predicted level to ascertain the blending efficacy and quantify the correction required to obtain the true mole fraction in the gas blend. This allowed accurate $[C_2H_2]$ standards to be derived, despite systematic uncertainties in the blending method, by conducting the same dilution steps but with the acetylene calibration cylinder in place of the methane calibration cylinder. This yielded a linear calibration fit with a multiplicative gain factor with which to correct $[C_2H_2]_r$ of 0.94, although this is only valid above 1.16 ppb $[C_2H_2]$, below which unstable $[C_2H_2]_r$ measurements were observed. As an additional test, a 0.24 Hz $\sigma_A$ of ±0.0863 ppb was derived at approximately 10 ppb $[C_2H_2]$, which represents the amount by which $[C_2H_2]$ is expected to vary between each consecutive measurement. The emphasises the stability of the Picarro G2203. The protocols used here can be applied to other gas analysers, especially in the absence of reference gas standards.

The same Picarro G2203 gas analyser was used to sample downwind of a controlled acetylene release alongside a LI-COR LI7810 gas analyser measuring $[CH_4]$. The acetylene release was conducted from within an active landfill site which was emitting methane. A $Q_{acetylene}$ of $0.242\,\mathrm{g\,s^{-1}}$ was measured by an acetylene flow meter, using a logging computer to automatically record $Q_{acetylene}$. A vehicle conducted twenty downwind transects of which fourteen could be used; six transects recorded features containing $[C_2H_2]_r$ measurements of less than −0.5 ppb in response to a sharp $[C_2H_2]$ increase, which the Picarro G2203 probably failed to spectrally fit on first instance. Calibrated Picarro G2203 $[C_2H_2]$ measurements were used in combination with corresponding LI-COR LI-7810 $[CH_4]$ measurements (interpolated to the lower Picarro G2203 $[C_2H_2]$ timestamp) to derive a landfill methane emission flux. As all $[C_2H_2]_r$ measurements of below 1.38 ppb were fixed to 0 ppb $[C_2H_2]$ for the calibrated dataset, a proportion of $[CH_4]$ measurements was also fixed to $[CH_4]_0$ to avoid overall flux bias during $Q_{methane}$ calculations.



The average landfill $Q_{\text{methane}}$ derived using calibrated Picarro G2203 $[C_2H_2]$ measurements was $17.3\,\text{g s}^{-1}$, with a standard deviation variability between the fourteen successful flux estimates of $\pm 9.6\,\text{g s}^{-1}$. This large variability can be attributed to the positioning of the source and downwind sampling positioning. The acetylene release point was not perfectly co-located with the centre of the methane source; successfully achieving this is challenging in a complex environment such a landfill site. This manifested itself through methane and acetylene plumes which that did not perfectly overlap on a downwind sampling plane. Another factor was the slow $[C_2H_2]_r$ sampling frequency, with gaps of up to $13\,\text{s}$ leading to information of the downwind sampling plumes being lost. A key outcome from this study was the derivation of fluxes using uncalibrated raw Picarro G2203 $[C_2H_2]_r$ measurements in place of calibrated $[C_2H_2]$, which yielded a lower $Q_{\text{methane}}$ of $(16.0\pm 8.6)\,\text{g s}^{-1}$. Although this 7.6% underestimation (compared to $Q_{\text{methane}}$ derived using calibrated $[C_2H_2]$) is smaller than the variability between individual flux estimates, it is possible to reduce the 56% random flux variability with improvements in the acetylene release methodology, for example by optimising the release location, with improved downwind sampling, sampling during better winds, conducting multiple acetylene releases or releasing acetylene at a higher rate. By contrast, the bias induced due to a lack of calibration is persistent and cannot be reduced by changing field sampling conditions. This therefore emphasises the importance of calibrating acetylene gas analysers used to derive tracer-based methane fluxes. Failure to do so could result in persistent biases in tracer-based flux estimates and hence, a biased understanding of the contribution of facility scale methane sources towards the overall global methane budget.

## Data availability

All data during laboratory testing and the field campaign will be published on a public repository upon final publication of this manuscript.

## Author contributions

AS conducted laboratory characterisation testing of the Picarro G2203, with support from OL, LL and CP. CY and OL arranged the supply of gas cylinders. AS and OL devised the acetylene calibration methodology. AS, TD and ML conducted the field sampling campaign. EE, CR and MT arranged for field site access. AS designed all field logging systems. AS and PK devised the emission flux quantification methodology, with support from GB. AS conducted all data analysis. AS wrote an initial draft of the manuscript which was revised by GB, OL, PC, PK, CY, TD and EE. PC, OL, GB, EE and CR organised funding for this work.

## Competing interests

The authors declare that they have no conflict of interest.



**Acknowledgements**

We thank the landfill site staff for facilitating site access, preparing the acetylene release location and for general support during the acetylene release campaign.

**Financial Support**

This work received funding from the Integrated Carbon Observation System National Network France. This work also received contributions in kind from SUEZ and the Chaire Industrielle TRACE, which is co-funded by the Agence Nationale de la Recherche (ANR) French National Research Agency (grant number: ANR-17-CHIN-0004-01), SUEZ, TotalEnergies - OneTech and Thales Alenia Space.

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
