# Peer review of "Accuracy of tracer-based methane flux quantification: underlying impact of calibrating acetylene measurements"

_EGUsphere, 2024_

## Author Comment (AC1)

Responses to individual reviewer comments (black text) are given below, in red text.

**Summary:**

This is a more impactful paper than the current title implies (see specific comments below). The study describes sources of systematic error affecting methane flux estimation using the tracer dispersion method, with acetylene as the tracer. It considers water vapour and other sources of spectroscopic/retrieval error, but the true novelty in the paper is most captured by the potential flux error that arises from non-calibration of acetylene analysers when used as a proxy for CH4. While the study is specific to the Picarro C2H2 analyser, the general need remains the same. The paper arrives at one flux bias number (7.6% - I think there's more to offer than this though), which is not insignificant in CH4 emissions quantification methods. Calibration such as that described here is rarely used by those using tracer methods for survey work, but clearly should be.

We thank the reviewer for taking the time to review our manuscript and for their valuable comments, which have helped to make significant improvements to the manuscript.

The paper is an excellent fit to AMT and its readers. The paper has value to scientists and industry using Picarro (and other IR) analysers and those following methane emissions quantification methods and datasets. It is a well-presented, rigorous, and detailed study. While it is refreshing to see such a detailed, lengthy, and complete technical study such as this, some of the important conclusions of the paper get lost at present (see specific comments).

We are grateful to the reviewer for their confidence in our research. We appreciate the time the reviewer has taken to understand the fundamental outcomes of this research and its overall importance. We have made every effort to address their concerns. We agree that the emphasis of key outcomes could be improved, which we address through several modifications to the manuscript, as discussed in the responses below.

I wholeheartedly recommend publication, and the comments below are constructive in the hope that the potential of the study can be better realised. I also have some technical questions which may need to be addressed satisfactorily.

We thank the reviewer for their kind recommendation for publication. We hope we have addressed all of the reviewer's questions in the responses below.

**Specific comments**

1.  Title: When I first saw the title, I thought this paper was about the simple calibration of an analyser. It is a fair bit more than this. It is about how calibration of an instrument significantly affects flux calculations. I would recommend a better title to attract readers to this important fact. It would be good to be explicit in the title that this is about impact on accuracy of fluxes.

We fully agree with the viewpoint of the reviewer. We have renamed the article to "Accuracy of tracer-based methane flux quantification: underlying impact of calibrating acetylene measurements". We hope this now emphasises the implications of failing to calibrate tracer gas measurements on methane fluxes.

2.  Introduction: I'm really not sure the historical discussion of the discovery of the greenhouse effect is needed (with very old references in places). It is not explicitly

linked to the content of the paper. Also, CH4 is the second-most important GHG according to the IPCC, after CO2 (not the third) – please correct, or otherwise clarify. If you are referring to GWP rather than RF (or something else), please be clear. Line 42 cites Dlugokencky et al., 2011 – please use the most recent reference for this regular review from Dlugokencky).

The reviewer is correct to state that methane is the second-most important anthropogenic greenhouse gas, according to the IPCC. We now explicitly clarify that the original statement refers to greenhouse gases in terms of overall radiative forcing. We have included a more recent reference to Schmidt *et al.* (2010) to reflect this. We thank the reviewer for this advice, to avoid ambiguity.

We recognise that some of the references are old. We have now added some more recent references to this introductory paragraph. For example, another citation to the latest IPCC report is made when discussing the effective radiative forcing of methane. In addition, we have now cited Saunois *et al.* (2024), which was not accepted for publication prior to submission of this manuscript. While we have included some of the latest publications to bring this section up to date, we would also prefer to maintain some older references to emphasise the prolonged nature of the issue of quantifying methane emissions. For example, in the same location as our citation to Dlugokencky *et al.* (2011), we also cited a more recent work from Lan *et al.* (2021), in the original manuscript. In this specific example there is a benefit to include references from different dates as this shows that uncertainties in the global methane budget has been a long and persistent problem. Hence, this fuels the motivation for the remainder of the manuscript.

Regarding the request to include the latest citation from Dlugokencky, it is not clear to us which specific work the reviewer is referring to here. We have already included the review from Lan *et al.* (2021). We have now included two additional review papers from the year 2019 and 2021 by Nisbet *et al.*, with Dlugokencky as a co-author in both articles. Perhaps, our citation to Saunois *et al.* (2024) can also satisfy this requirement, with Dlugokencky as a co-author in this work as well. We hope these are sufficiently recent citations to co-authored works by Dlugokencky, but we are unable to find a more recent paper by Dlugokencky as first author.

3. Line 100 – N2O has a lifetime >100 years (greater than CH4). The sentence here implies that use of N2O may be less problematic than CH4 (calling N2O "finite lifetime"). All gases have a finite lifetime in any case. Please clarify and correct this misleading sentence.

The reviewer makes a very valid point here and we agree that this could be misleading. We have now provided the duration of the nitrous oxide lifetime in the text and removed the word "finite".

4. Acetylene calibration: Ok…. So C2H2 standards aren't readily available. I understand that. Instead, an empirical calibration of mass flow controllers was used to produce a diluted air matrix. To do this, certified CH4 standards were used to define the dilution characteristics of specific mass flow controllers. This sounds ok, but I do have some questions. 1/ Was the same MFC tested repeatedly (e.g. on different days, after switching on and off etc), and were the results always consistent? If not, can you be confident that there was no drift in the MFC rates when switching between CH4 and C2H2? 2/ Can you be certain that there are no biases between these 2 gases in terms of

the experimental setup, e.g. C2H2 may be more sticky than CH4 in sample lines – was flushing and equilibrium time used, or can you be sure there are no residual effects? 3/ When you take into account the precision of the CH4 standard (i.e. how well known the concentration of the CH4 reference cylinder is), how does that precision manifest as a percentage of the concentration of acetylene after dilution? And is that percentage similar to the gain factor calculated for C2H2? It would be worth including a discussion of this in the paper for completeness and the avoidance of doubt.

In response to the first concern, the behaviour of mass-flow controllers has indeed been observed to change with time, as stated in the manuscript. For this reason, absolutely no changes were made to any of the flow connections, except for swapping the methane calibration cylinder with the acetylene calibration cylinder. The mass-flow controllers were not powered off for the full duration of each calibration test (during both methane and acetylene enhancements). We now emphasise these points further in Section 2.4. We are confident that the mass-flow controllers did not drift during each calibration test as the same methane gas blend was sampled three times, with each blend sampled 5.75 hours apart. The average standard deviation between each of the three $[CH_4]$ averages measured by the Picarro G2401 was ±2 ppb (see end of Section 2.3), with $[CH_4]$ reaching up to 12 ppm. Both methane and acetylene enhancements were measured within a 48-hour window as previously stated in Section 2.4, making it unlikely for any significant drift to occur during this period, considering the null drift observed over 11.5 hours.

The reviewer raises an astute point concerning the effect of the laboratory set-up on acetylene. The majority of gas during testing passed through Synflex 1300 tubing. Therefore, we have now conducted an additional $[C_2H_2]_r$ test by alternating the same gas stream, passing through either pure stainless steel or Synflex 1300. The average measured $[C_2H_2]_r$ was compared between the two gas streams and found to be identical (within uncertainty). This therefore confirms that Synflex 1300 has no noticeable effect on $[C_2H_2]$, compared to stainless steel. It is highly unlikely that Synflex 1300 and stainless steel have the exact same impact on acetylene. It can therefore be assumed that Synflex 1300 has no net effect on acetylene, allowing for its use during laboratory testing. The results of this new analysis are presented in Section S2 in the Supplement, with the outcome of the analysis summarised at the start of Section 2.1, when presenting the use of Synflex 1300 for the first time. We thank the reviewer for motivating us to include this very useful analysis.

The final point is an interesting one, regarding the declared $[CH_4]$ level in the methane calibration cylinder. We thank the reviewer for noticing this, due to their robust review of our methods. As this $[CH_4]$ level could not be accurately measured for verification (due to the argon content of this cylinder), all $[C_2H_2]_t$ (and hence all $C_{MFC}$) values rely on the accuracy of the declared $[CH_4]$ level of the methane calibration cylinder, which has an uncertainty of ±0.5%. Therefore, a new sensitivity analysis was conducted on the importance of this uncertainty range; a range of $C_{MFC}$ values was calculated using the range in possible $[C_2H_2]_t$ values (at each MFC setting), taking into account the uncertainty range in the declared $[CH_4]$ of the methane calibration cylinder. Rather than simply testing the effect of the variation in $C_{MFC}$ on deriving $[C_2H_2]$ standards (used to calibrate Picarro G2203 $[C_2H_2]_r$ measurements), we decided to also include the effect of the ±3% uncertainty associated with the declared $[C_2H_2]$ level of the acetylene calibration cylinder. This yielded a range in $[C_2H_2]_t$ levels (at each MFC setting) which were combined with corresponding $C_{MFC}$ values to obtain $[C_2H_2]$ reference standards. This analysis is presented in Section S5 in the Supplement, where Picarro G2203 $[C_2H_2]_r$ calibration fits are presented by both maximising and minimising $[C_2H_2]$ standards. This results in a possible extreme gain factor range of between 0.911 and 0.977,

which is presented in Section 2.4. It is important to note that this means that even in a worst-case scenario, the acetylene calibration gain factor is still significantly less than 1. Therefore, an acetylene calibration remains essential.

5. Flux error: A single number is given from the field trials of 7.6%. But a percentage isn't really that useful to those reading the paper, as a percentage of the flux will depend on all the factors you list that affect the quality of tracer release experiments and fluxes derived from them. A reader taking away the 7.6% number and using it as a guide to the effect of calibration would not be remembering something useful. However, I think you have all that you need here to derive a range of biases for different CH4 fluxes and acetylene release rates. They're all linearly scalable. So, rather than 7.6%, would it not be better to include a table in the paper that describes CH4 mass flux (g/s) error (and percent error), for a range of different assumed CH4 emission fluxes, or even simply C2H2/CH4 concentration ratios? This would be far more meaningful, and a better guide to others assessing how important calibration may be to their surveys. If you don't take up this recommendation, the discussion needs to be very explicit that the 7.6% number is not indicative of mass flux error due to instrument calibration and is unique to the conditions of this study (reducing the paper's impact).

We feel that the reviewer reached a different conclusion regarding the significance of this bias value, compared to the actual message we intended to convey. Perhaps this is because the original version of the manuscript lacked some discussion on the significance and origin of the bias value, which we now seek to rectify. This 7.6% value represents a bias that is largely independent of the sampling circumstances themselves including issues with sensor placement or downwind sampling location. Meanwhile, there is a ±56% random uncertainty between flux estimates which is unrelated to the acetylene calibration. This random uncertainty is indeed specific to the sampling campaign, which is discussed in depth in Section 4.2. However, this ±56% value is not the focus of this paper.

The 7.6% bias emphasised in the abstract and conclusion is principally due to the acetylene calibration. An acetylene calibration gain factor of 0.94 should result in a purely theoretical flux bias using uncalibrated measurements of exactly 6%, following eq. (4). However, this number is not exactly 6% for a number of reasons. The calibration was not valid below 1.19 ppb, meaning that all such $[C_2H_2]$ measurements were fixed to 0 ppb. To account for this, a threshold methane mole fraction was also used, below which all $[CH_4]$ measurements were set to the background methane mole fraction. Finally, the calibrated $[C_2H_2]$ data included a small linear offset which is only valid above 1.38 ppb $[C_2H_2]_r$. These effects collectively cause flux bias to vary from exactly 6%.

The reviewer is therefore technically correct to state that this 7.6% bias is specific to this study. A higher number of $[C_2H_2]_r$ measurements below 1.19 ppb from different transects will result in more $[C_2H_2]$ values being fixed to 0 ppb. Alongside this, the specific location and magnitude of $[CH_4]$ and $[C_2H_2]$ measurements from each transect effects the $[CH_4]$ threshold, designed to correct for the aforementioned effect. Hence, the magnitude of flux underestimation quantified for each transect using uncalibrated measurements instead of calibrated measurements varies. But by taking an average of fourteen different transects, individual variability in flux underestimation should average out. Therefore, although this 7.6% bias is specific to this study, we still believe it has merit, as it illustrates a typical order of magnitude level of flux bias induced by using uncalibrated measurements from a real-world sampling campaign, using the Picarro G2203 for tracer-based methane fluxes.

To include these points in the manuscript, additional extensive discussion has been included towards the end of Section 4.2. We hope this clarifies any doubt regarding the significance of this 7.6% flux underestimation using uncalibrated $[C_2H_2]_r$ measurements. We thank the reviewer for raising this, thus enabling us to thoroughly clarify this point.

6. There are a lot of important points to take from this paper, but they get lost in the extremely thorough discussion (and the conclusions). I would recommend completely changing the conclusions section to avoid unnecessary repetition of methods and instead focus on the salient conclusions about water vapour, need for acetylene calibration, and potential for flux error. It may also be useful to contextualise the error potential in terms of the magnitude of error from other flux methods (i.e. is it a comparable error if this calibration is ignored?). The important aspects of this useful paper will otherwise get missed.

We thank the reviewer for recognising the useful points in this paper. We agree that the key points of the paper may get lost in the conclusion in its original form. We have therefore rewritten the majority of the conclusion section and shortened it to place greater emphasis on the key outcomes of this work, with less focus on the methods. For example, details of the acetylene flow meter have now been omitted, with only the $Q_{acetylene}$ provided.

We have removed any discussion from Section 5 on the cause of the ±56% random uncertainty as it is a distraction and not the main focus of this work. Issues in the tracer flux method itself are also not the key outcome of this study. We thank the reviewer for their feedback, allowing us to place greater emphasis on the importance of the 8% bias, which is the main result, induced by failure to calibrate the Picarro G2203.

Regarding the suggestion to compare different flux uncertainties, in our opinion, the 8% uncertainty bias presented in this manuscript is not directly comparable to random uncertainties from other flux methods; a bias systematically forces the flux in one direction whereas a random uncertainty averages out over a sufficient number of samples. Furthermore, comparing flux errors from various flux methods is not the objective of this study, which will be the subject of a forthcoming study, as previously stated at the start of Section 4.1.

**Technical comments:**

1/ Line 152 – Full stop after bracket.

We thank the reviewer for spotting this oversight, which we have now rectified.

---

## Author Comment (AC2)

Responses to individual reviewer comments (black text) are given below, in red text.

The manuscript entitled "Measuring acetylene with a cavity ring-down spectroscopy gas

analyser and its use as a tracer to quantify methane emissions" by Shah et al. highlights the necessity of careful calibration when apply a commercial instrument measuring acetylene in quantifying methane emission flux. It provides valuable insight on the uncertainty of methane emission flux estimation which could be easily overlooked. The manuscript is well organized and presents instructive information on the experimental setup. I think the it well fits the scope of AMT journal and recommend the publication after minor revision. The following comments shall be considered during the revision.

We thank the reviewer for taking the time to thoroughly review this manuscript and for their useful comments. We thank the reviewer for spotting some key details, due to their meticulous review of our work. We address their individual comments below.

Compared to the major message (i.e., systematic bias can originate from uncareful calibration of the PICARRO instrument) the authors want to deliver, some detailed description on the experiment seems redundant in the main text. For example, discussion on the Allen variance obtained from zero air measurement, description on safety control of C2H2 release, etc.. I suggest the authors move those parts which are not directly relevant on the major message to Supplementary Materials, and revise the manuscript in a more concise way.

This is a useful suggestion. We have now moved details on the Allan variance test conducted at $[C_2H_2]$ a 0 ppb to the supplementary material, as it has little direct relevance for the rest of the manuscript. We have also moved information on the safety exclusion zone to the supplement. However, we have maintained some details on the acetylene release equipment as it adds value to the completeness of the manuscript, though we have made efforts to further reduce the size of this section by making the text more concise.

The PICARRO G2303 instrument used in this study showed 6% bias of measured C2H2 concentration. This results in on average 7.6% bias in estimating CH4 emission. It is well known that the sensitivity of optical instrument could drift with time. Therefore, the 6% bias could be largely instrument dependent. I suggest the authors specify this point. Moreover, the authors applied a complicated dilution system for their calibration. Detailed correction for the bias introduced by dilution is thus needed. The authors have described the reason of using high concentration C2H2. Does it mean that the calibration of PICARRO G2303 shall all following the same way?

The reviewer is totally correct to highlight that the 0.94 gain factor is specific to the Picarro G2203 instrument tested in this study. Calibration coefficients may vary for a different instrument used in a different study. We now emphasise this point in Section 2, when discussing the acetylene calibration fit. We also include this point more explicitly at the end of Section 4.2, when discussing the implications of using raw mole fraction measurements on calculated flux bias.

Regarding the topic of sensor drift, cavity ring-down spectroscopy is typically one of the most stable mole fraction techniques for measuring greenhouse gases. Previous studies on methane and carbon dioxide mole fraction have shown minimal changes in calibration factors over prolonged time periods. Based on this, we assume that calibration factors for $[C_2H_2]_r$ measurements can also remain valid over prolonged periods, in the context of $[C_2H_2]$

enhancements expected during sampling downwind of an acetylene release. Unfortunately, the long-term stability of $[C_2H_2]_r$ measurements has never been evaluated (to our knowledge), which would be an interesting subject of future research. This point has now been included in Section 2.4, where the calibration coefficients are presented. Some further discussion has also been added in Section 4.2.

Our dilution approach eliminates the effects of biases in MFC flow rates due to the use of $C_{MFC}$ factors, which correct $[C_2H_2]_t$ to obtain reference $[C_2H_2]$ levels. These $C_{MFC}$ factors were derived by comparing $[CH_4]_t$ to accurate $[CH_4]$ measurements made by the Picarro G2401 reference gas analyser. This dilution method is incredibly stable and does not drift over time as repeating the same dilution procedure three times, with gaps of 5.75 hours, resulted in average $[CH_4]$ values with a standard deviation variability of $(\pm2\pm1)$ ppb, as previously presented in the manuscript. However, it is possible there is an overall uncertainty associated with $[C_2H_2]_r$ calibration, as $[C_2H_2]_t$ and $[CH_4]_t$ rely on declared calibration cylinder mole fractions, with uncertainty ranges of $\pm3\%$ and $\pm0.5\%$, respectively. The influence of this is now investigated in Section S5 in the Supplement, by deriving calibration coefficients by propagating uncertainties from the extremities of the calibration cylinder mole fraction uncertainty ranges.

Line 169: Why applying such a measurement cycle?

This measurement cycle was not applied, but was beyond our control. The Picarro G2203 automatically provided measurements following this cycle. The possible reason for this is that the Picarro G2203 acquires a full acetylene absorbance spectrum to characterise baseline absorbance and then obtains a shorter reduced spectrum around the acetylene peak maximum. We have now clarified this point in the manuscript.

Line 170: "… are defined as wet measurement". This is confusing since the field measurements were conducted in dry conditions.

This is an interesting point and we understand the potential confusion. The point being made here is that all such measurements are raw, with no water-correction applied. These measurements include the effect of water when water is present. We have now clarified this in the manuscript and replaced the word "wet" with "raw".

Line 245: There are biased between the red dots and the gray lines under dry conditions in panel (a). Is this caused by the error introduced during dilution? Better provide a short statement in the following paragraph.

The reviewer makes an interesting observation that when sampling dry gas blends, $[C_2H_2]_t$ is consistently lower than $[C_2H_2]_r$. This is likely due to MFC offsets during dilution. However, this may also be related to the lack of calibration. As a calibration has not been conducted at this stage in the manuscript, we now present both options as possible causes of this disparity. Following the reviewer's recommendation, we now add a short discussion in the paragraph following this Figure 3. This is a very useful suggestion, as it now makes it clear that this analysis can only be used to compare empirical measurements.

Line 273: "6%". Should it be "6 ppb"?

We thank the reviewer for spotting this oversight, which we have now corrected.

Line 276: "[H2O]". Should it be "[C2H2]r"?

In fact, this refers to erroneous $[H_2O]_r$ spikes and not $[C_2H_2]_r$ spikes, measured by the Picarro G2203 gas analyser. To make this clearer, we have reorganised this sentence and now explicitly direct the reader to Figure 3 (b), where this phenomenon can be observed.

Line 291: What dose [C2H2]t stand for?

This stands for "targeted acetylene mole fraction", which is already defined at the start of Section 2.1. We hope that this previous definition is sufficient. In addition, we have now included the text "according to the MFC" in brackets in this location, to avoid confusion.

Line 432: "ppm" should be "ppb".

Yes, this is correct. We thank the reviewer for spotting this.

Line 439: "shorter" should be "longer".

We thank the reviewer from spotting this error.

Line 508: What is the placement height of the anemometer?

We were not able to make an accurate measurement of the anemometer height, as it dropped over time due to issues with the mast. However, we approximate a height above ground level of approximately 6 m. But in any case, this is irrelevant for flux computations using the tracer-based flux method.

Line 524: What is the sampling height? Would the height influence the large fluctuation showing in Figure 13?

Although we did not record the precise height above ground level of the air inlet during the campaign, we approximate this to be 2 m, based on the height of the vehicle.

The influence of sampling height on tracer-based methane flux variability is expected to be minimal, with the boundaries of the landfill site being approximately 300 m away from the sampling road. Sampling took place in a sufficiently flat area with no large hills or obstacles to cause topographical variation between the emission source and the point of measurement, allowing for a smooth view of the plume. However, it is true that the methane and acetylene plumes were offset as the acetylene release was not perfectly co-located with the landfill methane source, with methane emissions taking place unevenly over a large diffuse area. As a result, each gas may experience a slightly different transport path under different turbulent patterns between the point of emission and the point of measurement, leading to spatial or temporal separation in the plumes as they move downwind. Although near-surface turbulence could explain part of the variability shown in Figure 13, the turbulent conditions were approximately the same for the methane and acetylene plumes. Therefore, this variability is not exacerbated by sampling at 2 m above ground level, as opposed to sampling from some different sampling height.